# Noninvertible symmetry and topological holography for modulated SPT in one dimension

Jintae Kim,[1, 2, *] Yizhi You,[3, †] and Jung Hoon Han[1, ‡]

[1]*Department of Physics, Sungkyunkwan University, Suwon 16419, South Korea*
[2]*Institute of Basic Science, Sungkyunkwan University, Suwon 16419, South Korea*
[3]*Department of Physics, Northeastern University, 360 Huntington Ave, Boston, MA 02115, USA*
(Dated: September 25, 2025)

We examine noninvertible symmetry (NIS) in one-dimensional (1D) symmetry-protected topological (SPT) phases protected by dipolar and exponential-charge symmetries, which are two key examples of modulated SPT (MSPT). To set the stage, we first study NIS in the $\mathbb{Z}_N \times \mathbb{Z}_N$ cluster model, extending previous work on the $\mathbb{Z}_2 \times \mathbb{Z}_2$ case. For each symmetry type (charge, dipole, exponential), we explicitly construct the noninvertible Kramers-Wannier (KW) and Kennedy-Tasaki (KT) transformations, revealing dual models with spontaneous symmetry breaking (SSB). The resulting symmetry group structure of the SSB model is rich enough that it allows the identification of other SSB models with the same symmetry. Using these alternative SSB models and KT duality, we generate novel MSPT phases distinct from those associated with the standard decorated domain wall picture, and confirm their distinctiveness by projective symmetry analyses at their interfaces. Additionally, we establish a topological-holographic correspondence by identifying the 2D bulk theories-two coupled layers of toric codes (charge), anisotropic dipolar toric codes (dipole), and exponentially modulated toric codes (exponential)-whose boundaries host the respective 1D MSPT phases.

## CONTENTS

## I. INTRODUCTION

For lattice many-body models, it is customary to express global symmetry ($S$) as products of on-site symmetries ($S_j$) in the form $S = \prod_j S_j$, where $j$ are the sites of the lattice. The concept of symmetry can be generalized to include modulation by rigorously defining the symmetry group $G$ as [1–7]

$$G = G_{\text{int}} \rtimes G_{\text{sp}} \qquad (1.1)$$

where $G_{\text{int}}$ denotes the internal symmetry group and $G_{\text{sp}}$ represents the spatial symmetry group. Here, the semidirect product indicates that an element of $G_{\text{int}}$ is non-trivially transformed under the action of an element of $G_{\text{sp}}$, i.e., $g_{\text{sp}} g_{\text{int}} g_{\text{sp}}^{-1} \neq g_{\text{int}}$ where $g_{\text{int}} \in G_{\text{int}}$, $g_{\text{sp}} \in G_{\text{sp}}$. If all elements of $G_{\text{int}}$ are instead transformed trivially by elements of $G_{\text{sp}}$, the resulting symmetries are non-modulated and the symmetry group $G$ is expressed as $G = G_{\text{int}} \times G_{\text{sp}}$.

When considering translational symmetry, which is the main focus in this paper, the examples of modulated symmetries are symmetries with each $S_j$ raised to position-dependent function $f_j$, and the global symmetry is expressed as $S = \prod_j (S_j)^{f_j}$. Local symmetry charges are spatially modulated and take on the position-dependent value $f_j$. Typical examples of modulated symmetries are dipole ($f_j \propto j$), quadrupole

* Electronic address: jint1054@gmail.com
† Electronic address: y.you@northeastern.edu
‡ Electronic address: hanjemme@gmail.com

($f_j \propto j^2$), and other multipolar symmetries. Even an exponentially modulated charge symmetries with $f_j \sim a^j$ for some integer $a > 1$ is possible. All of these examples undergo nontrivial transformations under translational symmetry.

Various one-dimensional (1D) SPT models protected by modulated symmetries have been proposed and analyzed [2–4, 8]. Modulated symmetries represent one form of extension of the conventional notion of global symmetry, along with higher-form [9, 10] and subsystem symmetries [11, 12] that have gained much attention in recent years.

Another notable extension of the symmetry concept in the form of noninvertible symmetry (NIS) has taken place and is being pursued with vigor - see [13–15] for recent reviews. For lattice models, it is well appreciated by now that familiar transformations such as the Jordan-Wigner [16], Kramers-Wannier (KW) [17], and Kennedy-Tasaki (KT) [18, 19] transformations are, in fact, noninvertible operations when performed on a periodic chain. A recent interesting perspective is to view these NIS operations through the lens of topological holography where these algebraic transformations correspond to certain geometric changes in the boundary conditions at the edges of two-dimensional (2D) topological models [20, 21]. A particularly intriguing recent advance is the observation that the cluster model - a toy example of $\mathbb{Z}_2 \times \mathbb{Z}_2$ symmetry-protected phase in 1D - is protected by a third, KW symmetry which is noninvertible [22]. This extra noninvertible symmetry suggests the notion of the phase being protected by fusion category symmetry rather than group-based symmetry, and it is remarkable that this more abstract form of SPT [23–25] can be realized in such a simple model. Certain non-Abelian extensions of the cluster model with noninvertible symmetries have also been explored [25–27].

Motivated by these advances, we ask whether SPT phases protected by modulated symmetries may also exhibit noninvertible symmetries, potentially leading to intriguing consequences. For convenience we refer to modulated SPTs as *MSPTs*, and those with an additional noninvertible symmetry as *NIMSPTs*. Our investigation shows that existing MSPTs, at least those considered in our work, are also NIMSPTs. To be precise, an existing MSPT is one example of NIMSPT among a larger set of SPTs protected by modulated symmetries as well as one noninvertible symmetry. We thus enlarge the possible types of topological matter in 1D to include these various NIMSPTs. In addition, we explore whether the concept of topological holography - a broad paradigm relating $d$-dimensional topological models with certain symmetry to $(d + 1)$-dimensional models whose symmetry is inherited by the boundary modes [20, 21, 28–32] - can be applicable in understanding various NIMSPT phases. In each case of MSPT we successfully identify the corresponding bulk Hamiltonian and prove that its boundary mode is indeed the NIMSPT with a given modulated symmetry. The noninvertible symmetry operation of each NIMSPT can be understood as changes in the boundary conditions of the corresponding bulk topological model.

Our discussion begins by generalizing the investigation of NIS and its implications for the $\mathbb{Z}_2 \times \mathbb{Z}_2$ cluster model [22] to arbitrary $\mathbb{Z}_N \times \mathbb{Z}_N$ cluster model [33]. Such groundwork

provides the framework and many of the techniques needed for treating the more challenging NIMSPTs with $\mathbb{Z}_N$ dipole symmetry and exponential charge symmetries. To distinguish several different SPTs treated in this work, we refer to the SPT embodied in the $\mathbb{Z}_N \times \mathbb{Z}_N$ cluster model as the 'charge' SPT or cSPT, reflecting the fact that this SPT phase is being protected by a pair of on-site, uniform (rather than modulated) symmetries. We then move to discuss the dipolar SPT (dSPT) phase embodied in a dipolar cluster state protected by two charge and two dipole symmetries and identify the NIS associated with it. The dipolar cluster model written down and analyzed in an earlier work [2, 3, 34] has only one charge and one dipole symmetry protecting it. The new dipolar cluster model we introduce and analyze here, on the other hand, turns out to provide a more natural setting for the discussion of NIS. Finally, we explore the noninvertible symmetry and its consequences of the exponential cluster model [2] embodying the exponential SPT (eSPT).

Major outcomes of our investigations can be summarized as follows. For the cSPT (Sec. II), we identify two new families of cSPTs besides the cluster states by generalizing the roadmap laid out in [22]. Each family is parameterized by an integer $\alpha \in \mathbb{Z}_N$ and represents a distinct SPT different from the cluster state when $\alpha \neq 0$ and from each other when $\alpha \neq \alpha'$. The original cSPT itself is shown to be the boundary theory of *two coupled layers* of $\mathbb{Z}_N$ toric codes. For the dSPT (Sec. III), we first write down a new cluster model with dipole symmetries and identify the appropriate KW and KT transformation. By an extension of the roadmap developed for cSPT, we identify a family of new dSPTs parameterized by two integers $\alpha, \beta \in \mathbb{Z}_N$. The dSPT is shown to be the boundary theory of two coupled layers of anisotropic, dipolar toric code Hamiltonian proposed in [35]. For the eSPT (Sec. IV) we perform the similar analysis and arrive at a new family of eSPTs parameterized by $\alpha \in \mathbb{Z}_N$. The exponential cluster model appears as the boundary theory of two coupled layers of exponentially modulated toric codes proposed in [36, 37]. In all three cases, the corresponding KW transformation can be interpreted as changes in the boundary conditions from rough to smooth boundaries of the appropriately chosen 2D bulk models. For ease of readership, a summary of the distinct NIMSPT Hamiltonians, highlighting their structural and physical characteristics, is provided in Sec. V A.

Overall, this work represents a meaningful addition to the growing list of literature on lattice model with NIS [38–48], as well as the burgeoning field of 1D SPTs protected by modulated symmetries [2–4, 8]. Particular emphases are given to the marriage of modulated symmetry, noninvertible symmetry, and topological holography through investigation of several explicit models of 1D MSPT.

## II. CHARGE SYMMETRY

The $\mathbb{Z}_N$ qudit degrees of freedom are represented by $|g_j\rangle_j$, with $g_j \in \mathbb{Z}_N$, at each site $j$ of a 1D chain. The $\mathbb{Z}_N$ Pauli

operators $(Z,X)$ act on these states as

$$Z_j|g_j\rangle_j = \omega^{g_j}|g_j\rangle_j, \qquad X_j|g_j\rangle_j = |g_j+1\rangle_j, \qquad (2.1)$$

where $\omega = \exp(2\pi i/N)$. Throughout the paper we use the notation

$$\mathscr{O} \xrightarrow{W} \mathscr{O}' \text{ for } W\mathscr{O} = \mathscr{O}'W, \qquad (2.2)$$

representing the conjugation of the operator $\mathscr{O}$ by $W$ which is either invertible or noninvertible.

This section covers the noninvertible symmetry associated with gauging the charge symmetry, investigate some well-known as well as new SPT phases protected by such noninvertible symmetry, and conclude with their holographic interpretation as the boundary theory of two coupled toric codes.

## A. Gauging and noninvertible charge symmetry

We consider the gauging of the global charge symmetry

$$C = \prod_j X_j \qquad (2.3)$$

for the matter fields $(X_j, Z_j)$ in 1D. This can be done by introducing gauge fields $(\bar{X}_j, \bar{Z}_j)$ at the site $j$ and the local gauge symmetry operator

$$g_j = X_j \bar{Z}_{j-1} \bar{Z}_j^\dagger. \qquad (2.4)$$

The global symmetry operator $C$ can be expressed as the product of $g_j$'s: $C = \prod_j g_j$.

The operators that commute with $C$ are $X_j$, $Z_j^\dagger Z_{j+1}$, which map under the gauging to gauge fields

$$X_j \to \bar{Z}_{j-1}^\dagger \bar{Z}_j, \qquad Z_j^\dagger Z_{j+1} \to \bar{X}_j. \qquad (2.5)$$

Formally, it can be implemented by the KW operator $K_{\text{KW}}$ [5, 19, 39]:

$$K_{\text{KW}} = \sum_{\mathbf{g},\mathbf{g}'} \omega^{\Sigma_j(g_{j-1}-g_j)g'_{j-1}}|\mathbf{g}'\rangle\langle\mathbf{g}|, \qquad (2.6)$$

where $|\mathbf{g}\rangle = \otimes_j|g_j\rangle_j$ and $|\mathbf{g}'\rangle = \otimes_j|g'_j\rangle_j$. One can show

$$X_j \xrightarrow{K_{\text{KW}}} Z_{j-1}^\dagger Z_j, \qquad Z_j^\dagger Z_{j+1} \xrightarrow{K_{\text{KW}}} X_j, \qquad (2.7)$$

where the bars have been removed on the right side of the arrow. The operator $K_{\text{KW}}$ is noninvertible, and can be expressed as the product of an invertible, unitary operator and a noninvertible projection operator [22, 41].

## B. Charge SPT

The 1D SPT phase protected by $\mathbb{Z}_N \times \mathbb{Z}_N$ symmetry is exemplified by the $\mathbb{Z}_N$ cluster model [49–51]

$$H_c = -\sum_j (Z_{2j-1}X_{2j}Z_{2j+1}^\dagger + Z_{2j-2}^\dagger X_{2j-1}Z_{2j}) + \text{h.c.}. \qquad (2.8)$$

The chain is assumed either infinite or when finite and periodic, to have the size $L$ even. The ground state of the cluster Hamiltonian is given by

$$|\psi_c\rangle \propto \sum_{\mathbf{g}} \omega^{\Sigma_j g_{2j}(g_{2j-1}-g_{2j+1})}|\mathbf{g}\rangle$$
$$= \sum_{\mathbf{g}} \omega^{\Sigma_j g_{2j-1}(g_{2j}-g_{2j-2})}|\mathbf{g}\rangle. \qquad (2.9)$$

Introducing the controlled-Z operation

$$\text{CZ}_{i,j}|g_i\rangle_i|g_j\rangle_j = \omega^{g_i g_j}|g_i\rangle_i|g_j\rangle_j,$$

the ground state $|\psi_c\rangle$ is obtained by the action of the unitary operator

$$U_c = \prod_j \text{CZ}_{2j,2j-1}\text{CZ}_{2j,2j+1}^\dagger$$
$$= \prod_j \text{CZ}_{2j-1,2j-2}^\dagger\text{CZ}_{2j-1,2j} \qquad (2.10)$$

on the product state $|+\rangle = \prod_j|+\rangle_j$ where $X_j|+\rangle_j = |+\rangle_j$:

$$|\psi_c\rangle = U_c|+\rangle.$$

It can be shown that

$$X_j \xrightarrow{\text{CZ}_{ij}} Z_i X_j, \qquad X_j \xrightarrow{\text{CZ}_{ij}^\dagger} Z_i^\dagger X_j$$

and thus $U_c$ implements

$$X_{2j} \xrightarrow{U_c} Z_{2j-1}X_{2j}Z_{2j+1}^\dagger,$$
$$X_{2j-1} \xrightarrow{U_c} Z_{2j-2}^\dagger X_{2j-1}Z_{2j}. \qquad (2.11)$$

This results in the mapping $-\sum_j(X_j+X_j^\dagger) \xrightarrow{U_c} H_c$.

The $\mathbb{Z}_N$ cluster model hosts two global charge symmetries:

$$C^o = \prod_j X_{2j-1}, \qquad C^e = \prod_j X_{2j}, \qquad (2.12)$$

over the odd and even sublattices, respectively. They are non-modulated symmetries in the sense that, when translational symmetry under two-site translation is considered, they remain invariant [52]. Throughout this paper, the translational symmetry we have in mind is the symmetry under two-site translation. Accordingly, the topological phase protected by a pair of charge symmetries will be referred to as charge SPT, or cSPT.

In addition, the cluster model exhibits a noninvertible symmetry characterized by the commutation $K_c H_c = H_c K_c$, where the noninvertible KW operator $K_c$ is defined as [5, 19, 22, 39]:

$$K_c = T K_{\text{KW}}^o K_{\text{KW}}^e. \qquad (2.13)$$

Here $K_{\text{KW}}^o$ and $K_{\text{KW}}^e$ are the KW operators acting on the odd and even sublattices, respectively, and

$$T = \sum_{\mathbf{g}} \bigotimes_j |g_j\rangle_{j+1}\langle g_j|_j$$

is the translation by one site. It performs

$$X_j \xrightarrow{K_c} Z_{j-1}^\dagger Z_{j+1}, \qquad Z_{j-1}^\dagger Z_{j+1} \xrightarrow{K_c} X_j, \qquad (2.14)$$

and therefore

$$Z_{2j-1} X_{2j} Z_{2j+1}^\dagger \xrightarrow{K_c} Z_{2j-1}^\dagger X_{2j}^\dagger Z_{2j+1},$$
$$Z_{2j-2}^\dagger X_{2j-1} Z_{2j} \xrightarrow{K_c} Z_{2j-2}^\dagger X_{2j-1} Z_{2j}. \qquad (2.15)$$

The first term $Z_{2j-1} X_{2j} Z_{2j+1}^\dagger$ is transformed into its Hermitian conjugate under $K_c$; the second term $Z_{2j-2}^\dagger X_{2j-1} Z_{2j}$ is transformed to itself. Overall, the cluster Hamiltonian $H_c$ is transformed to itself under $K_c$ [53]. One can check $K_c^\dagger = K_c$.

The two invertible and one noninvertible symmetry operators $\{C^o, C^e, K_c\}$ of the cluster model span the fusion algebra:

$$C^e K_c = K_c C^e = K_c, \qquad C^o K_c = K_c C^o = K_c,$$
$$(K_c)^\dagger K_c = (K_c)^2 = \left( \sum_{k=1}^N (C^e)^k \right) \left( \sum_{k=1}^N (C^o)^k \right). \qquad (2.16)$$

The first two relations imply that only the $C^o = +1$ and $C^e = +1$ eigenstates survive the projection inherent in $K_c$. The $K_c$ operator becomes $K_c^\dagger K_c = K_c^2 = N^2$ and is de facto invertible in such a *symmetric* sector. Applying the KW transformation twice gives back the original model, in accord with the usual notion of KW performing a duality operation.

The three symmetries $\{C^o, C^e, K_c\}$ of the cluster model do not form a group and are not subject to the group-based cohomology classification. A more general classification scheme for such fusion category symmetry has been formulated [14, 15, 23–25]. For $N = 2$, the fusion algebra spanned by $\{C^o, C^e, K_c\}$ is equivalent to that of the fusion category Rep($D_8$) [22]. For $N > 2$, the fusion algebra of (2.16) is that of the Tambara-Yamagami type, TY($\mathbb{Z}_N \times \mathbb{Z}_N$) [5]. In line with recent terminology in the physics literature, we refer to the SPT phase protected by fusion category symmetry as noninvertible SPT, or NISPT. The ultimate goal of this work is to generalize NISPT to SPT protected by modulated symmetries, or MSPTs, to achieve gross understanding of what we would call the NIMSPT.

### C. Kennedy-Tasaki transformation

The Kennedy-Tasaki (KT) transformation maps the SPT phase to the spontaneous symmetry breaking (SSB) phase and vice versa. For the cSPT it is implemented by [5, 19, 54, 55]

$$\mathrm{KT}_c = U_c K_c U_c^\dagger \qquad (2.17)$$

where $U_c$ defined in (2.10) maps the paramagnetic state $\sum_{\mathbf{g}} |\mathbf{g}\rangle$ to the cluster ground state. One can show

$$X_{2j-1} \xrightarrow{\mathrm{KT}_c} X_{2j-1}^\dagger,$$
$$X_{2j} \xrightarrow{\mathrm{KT}_c} X_{2j},$$
$$Z_{2j-1}^\dagger Z_{2j+1} \xrightarrow{\mathrm{KT}_c} Z_{2j-1} X_{2j} Z_{2j+1}^\dagger,$$
$$Z_{2j}^\dagger Z_{2j+2} \xrightarrow{\mathrm{KT}_c} Z_{2j}^\dagger X_{2j+1} Z_{2j+2}. \qquad (2.18)$$

Using $K_c^\dagger = K_c$, it follows that $(\mathrm{KT}_c)^\dagger = \mathrm{KT}_c$ and $(\mathrm{KT}_c)^2 = (K_c)^2$. The two charge symmetry operators of the cluster model transform under $\mathrm{KT}_c$ as

$$C^o \xrightarrow{\mathrm{KT}_c} (C^o)^\dagger, \qquad C^e \xrightarrow{\mathrm{KT}_c} C^e. \qquad (2.19)$$

On the other hand, $K_c$ under the KT conjugation becomes

$$\mathrm{KT}_c K_c = (U_c P^o) \mathrm{KT}_c \equiv V_c \cdot \mathrm{KT}_c, \qquad (2.20)$$

where $P^o = \prod_i P_{2i+1}$ is the product of on-site charge conjugation

$$P_j = \sum_{g_j} |-g_j\rangle_j \langle g_j|_j$$

on odd sites. For $N = 2$, $V_c = U_c P^o$ reduces to $U_c$ as $P^o$ becomes an identity, recovering the $N = 2$ expression obtained earlier [22].

Conjugating the cluster model $H_c$ by $\mathrm{KT}_c$ results in two copies of $\mathbb{Z}_N$ Ising model. First, the conjugation by $U_c^\dagger$ transforms $H_c$ to the trivial model $-\sum_j (X_j + X_j^\dagger)$. Performing KW on this gives, according to (2.14), the double Ising model

$$\hat{H}_c = -\sum_j Z_{j-1}^\dagger Z_{j+1} + \mathrm{h.c.}, \qquad (2.21)$$

which remains invariant under the final conjugation by $U_c$. It thus follows that

$$H_c \xrightarrow{\mathrm{KT}_c} \hat{H}_c,$$

and the reverse $\hat{H}_c \xrightarrow{\mathrm{KT}_c} H_c$ can be checked easily. The trivial SPT phase $-\sum_j (X_j + X_j^\dagger)$, on the other hand, is mapped to itself under $\mathrm{KT}_c$, not to the double Ising model.

Transformations of symmetry operators of the cluster model under $\mathrm{KT}_c$ are summarized as

$$\{C^o, C^e, K_c\} \xrightarrow{\mathrm{KT}_c} \{(C^o)^\dagger, C^e, V_c\}. \qquad (2.22)$$

Since $V_c^2 = U_c (P^o)^2 U_c^\dagger = U_c U_c^\dagger = 1$, the symmetry group generated by $V_c$ is $\mathbb{Z}_2$. For general $N$, $V_c$ has a non-trivial commutation relation with $C^o$, i.e. $V_c C^o = (C^o)^\dagger V_c$, and one can check that $C^o$ generates a normal subgroup in a group generated by $\{C^o, V_c\}$. The overall symmetry group spanned by $\{(C^o)^\dagger, C^e, V_c\}$ is characterized as

$$\mathbb{Z}_N^e \times (\mathbb{Z}_N^o \rtimes \mathbb{Z}_2^{V_c}), \qquad (2.23)$$

where $\mathbb{Z}_2^{V_c}$ acts as an outer automorphism on $\mathbb{Z}_N^o$. The double Ising model indeed possesses the symmetries of (2.23). Note that all three symmetries are now invertible, since $(V_c)^2 = 1$. The action by $K_c$ effectively performs the projection to the symmetric sector, and acting solely within this sector is what transforms the noninvertible $K_c$ into the invertible operator $V_c$.

The SSB ground states of the double Ising model are characterized by a pair of $\mathbb{Z}_N$ integers $|g^o, g^e\rangle$ representing the two charge SSB states on the even and odd sublattice, respectively:

$$g_{2j-1} = g^o, \quad g_{2j} = g^e.$$

The order parameters characterizing the SSB ground states are $\sum_j Z_{2j+1}$ and $\sum_j Z_{2j}$. Each ground state $|g^o, g^e\rangle$ generally breaks all three symmetries of the double Ising model, yet the ground state degeneracy (GSD) remains at $N^2$ rather than $2N^2$. For $N = 2$, this was attributed to the existence of a hidden $\mathbb{Z}_2$ symmetry group $\{1, V_c\}$ which leaves each ground state invariant [22]. For $N > 2$, however, such explanation will undergo some modification.

Let us choose a ground state $|g^o, g^e\rangle$ and act on it with elements of the symmetry group $(C^o)^{\eta_1}(C^e)^{\eta_2}(V_c)^{\eta_3}$, where $\eta_1, \eta_2 \in \mathbb{Z}_N$ and $\eta_3 \in \mathbb{Z}_2$:

$$(C^o)^{\eta_1}(C^e)^{\eta_2}(V_c)^{\eta_3}|g^o, g^e\rangle = |\eta_1 + (-1)^{\eta_3}g^o, \eta_2 + g^e\rangle. \quad (2.24)$$

It shows that any ground state generated by $V_c$ can alternatively be generated by $C^o$ raised to an appropriate power $\eta_1$. This explains why the GSD is still $N^2$.

A further consequence of (2.24) is that each ground state $|g^o, g^e\rangle$ is invariant under $(C^o)^{\eta_1}(C^e)^{\eta_2}(V_c)^{\eta_3}$ when the condition

$$\eta_1 = 2g^o \pmod{N}, \quad \eta_2 = 0, \quad \eta_3 = 1 \quad (2.25)$$

is satisfied. The invariant symmetry operation by $(C^o)^{\eta_1}V_c$ is $\mathbb{Z}_2$ since $[(C^o)^{\eta_1}V_c]^2 = 1$. When $N = 2$, $\eta_1 = 0$ and $\{1, V_c\}$ forms a $\mathbb{Z}_2$ subgroup whose action on any ground state yields identity. For general $N > 2$, however, $\eta_1$ varies with the quantum number $g^o$ of the ground state and it is not appropriate to view $\{1, (C^o)^{\eta_1}V_c\}$ as forming a subgroup. We will shortly introduce another model of SSB sharing the same symmetry group and the GSD as the double Ising model, but in which the structure of the invariant symmetry operation is quite different.

### D. Other cSPT states

This subsection is devoted to the discovery of a new SPT model sharing the same set of symmetries $\{C^o, C^e, K_c\}$ as the cluster model, while lying outside the conventional cohomology classification scheme based solely on the two charge symmetries. SPT phases protected by $C^o$ and $C^e$ are classified by the second cohomology $H^2(\mathbb{Z}_N \times \mathbb{Z}_N, U(1)) = \mathbb{Z}_N$, and represented by the cluster model

$$H_c^{(k)} = \sum_j (Z_{2j-1}^k X_{2j} Z_{2j+1}^{-k} + Z_{2j-2}^{-k} X_{2j-1} Z_{2j}^k) + \text{h.c.} \quad (2.26)$$

for $k \in \mathbb{Z}_N$. All the discussions of $K_c$ and $\text{KT}_c$ in the previous section can be generalized straightforwardly to arbitrary $k$ by introducing generalized KW operator $K_{\text{KW}}^{(k)}$

$$K_{\text{KW}}^{(k)} = \sum_{\mathbf{g}, \mathbf{g}'} \omega^{\sum_j k(g_{j-1} - g_j)g'_{j-1}} |\mathbf{g}'\rangle\langle\mathbf{g}|, \quad (2.27)$$

and defining

$$K_c^{(k)} = T(K_{\text{KW}}^{(k)})^o (K_{\text{KW}}^{(k)})^e, \quad \text{KT}_c = U_c K_c^{(k)} U_c^\dagger.$$

It turns out, however, that $K_c^{(k)} H_c \neq H_c K_c^{(k)}$ unless $k = 1$.

#### 1. Other cSPT states : $H_{c'}$

Now, we will construct another cSPT protected by $C^o$, $C^e$, and $K_c$. To this end we introduce the $\mathbb{Z}_N$ $Y$-operator defined as $Y = \omega^{1/2}XZ$ ($\omega^{1/2} = e^{\pi i/N}$). The algebra among $(X, Y, Z)$ is

$$ZX = \omega XZ, \quad ZY = \omega YZ, \quad YX = \omega XY, \quad (2.28)$$

recovering the well-known Pauli algebra and $Y = Y^\dagger$ at $N = 2$. This operator transforms under $V_c$ as

$$Y_{2j-1} \xrightarrow{V_c} Z_{2j-2}Y_{2j-1}^\dagger Z_{2j}^\dagger, \quad Y_{2j} \xrightarrow{V_c} Z_{2j-1}Y_{2j}Z_{2j+1}^\dagger,$$

$$Z_{2j-1} \xrightarrow{V_c} Z_{2j-1}^\dagger, \quad Z_{2j} \xrightarrow{V_c} Z_{2j}, \quad (2.29)$$

thus

$$Y_{2j-1}Y_{2j+1}^\dagger \xrightarrow{V_c} Z_{2j-2}Y_{2j-1}^\dagger Z_{2j}^{-2}Y_{2j+1}Z_{2j+2}, \quad (2.30)$$

making the sum $Y_{2j-1}Y_{2j+1}^\dagger + Z_{2j-2}Y_{2j-1}^\dagger Z_{2j}^{-2}Y_{2j+1}Z_{2j+2}$ invariant under $V_c$.

Based on this observation we write down a new model sharing the same $\{C^o, C^e, V_c\}$ symmetries as the double Ising model:

$$\hat{H}_{c'} = -\sum_j \omega^\alpha Z_{2j}Z_{2j+2}^\dagger$$
$$- \sum_j Y_{2j-1}Y_{2j+1}^\dagger (1 + Z_{2j-2}^\dagger Z_{2j}^2 Z_{2j+2}^\dagger) + \text{h.c.}, \quad (2.31)$$

where $\alpha \in \mathbb{Z}_N$ characterizes a distinct SSB [56]. All the terms in this Hamiltonian mutually commute. The first term is minimized by $Z_{2j}Z_{2j+2}^\dagger = \omega^{-\alpha}$, and the second term is reduced to $-Y_{2j-1}Y_{2j+1}^\dagger (1 + Z_{2j-2}Z_{2j}^{-2}Z_{2j+2}) \to -2Y_{2j-1}Y_{2j+1}^\dagger$ in the ground state. The overall ground state conditions are

$$Z_{2j}Z_{2j+2}^\dagger = \omega^{-\alpha}, \quad Y_{2j-1}Y_{2j+1}^\dagger = 1. \quad (2.32)$$

The first condition in (2.32) imposes $g_{2j+2} = g_{2j} + \alpha$ in the $Z$-basis at the even sites, giving $g_{2j} = g^e + \alpha j$ for some $g^e \in \mathbb{Z}_N$. The order parameter for this state is $\sum_j \omega^{-\alpha j}Z_{2j}$. The second condition imposes $g_{2j-1}^Y = g^Y$ for some $g^Y$ in the $Y$-basis $(Y|g^Y\rangle = \omega^{g^Y}|g^Y\rangle)$ at the odd sites. The order parameter for the odd-site state is $\sum_j Y_{2j-1}$. The overall SSB ground states are labeled by these two quantum number as $|(g^Y)^o, g^e\rangle$. Note that $\alpha$ is a fixed parameter of the Hamiltonian $H_{c'}$ and not a quantum number of the ground state.

One can show

$$\omega^{\eta_1}Z_{2j-2}Y_{2j-1}^\dagger Z_{2j}^\dagger \xrightarrow{(C^o)^{\eta_1}(C^e)^{\eta_2}V_c} Y_{2j-1}$$
$$\omega^{\eta_2}Z_{2j} \xrightarrow{(C^o)^{\eta_1}(C^e)^{\eta_2}V_c} Z_{2j}, \quad (2.33)$$

implying

$$(C^o)^{\eta_1}(C^e)^{\eta_2}V_c^{\eta_3}|(g^Y)^o, g^e\rangle$$
$$= |\eta_1 - \delta_{\eta_3,1}\alpha + (-1)^{\eta_3}(g^Y)^o, \eta_2 + g^e\rangle. \quad (2.34)$$

It follows that the ground state $|(g^Y)^o, g^e\rangle$ is invariant under $(C^o)^{\eta_1}(C^e)^{\eta_2}V_c^{\eta_3}$ when

$$\eta_1 = 2(g^Y)^o + \alpha \ (\mathrm{mod}\ N), \quad \eta_2 = 0, \quad \eta_3 = 1. \qquad (2.35)$$

Each SSB ground state of $\hat{H}_{c'}$ has one operator $(C^o)^{\eta_1}V_c$ with $\eta_1$ fixed by (2.35) that leaves it invariant. In general, $(C^o)^{\eta_1}V_c$ squares to one. For $N = 2$, $\eta_1 = \alpha$ is independent of the ground state and one has a state-preserving $\mathbb{Z}_2$ subgroup $\{1, (C^o)^{\eta_1}V_c\}$. For $N > 2$, the element $(C^o)^{\eta_1}V_c$ depends on $(g^Y)^o$ of the ground state via (2.35).

We proceed to apply $\mathrm{KT}_c^\dagger$ on the newly found SSB model $\hat{H}_{c'}$ to arrive at a new family of SPT models $\hat{H}_c \xrightarrow{\mathrm{KT}_c^\dagger} H_{c'}$ parameterized by $\alpha \in \mathbb{Z}_N$:

$$H_{c'} = -\sum_j \omega^{-\alpha} Z_{2j}^\dagger X_{2j+1} Z_{2j+2}$$
$$- \sum_j Y_{2j-1} X_{2j} Y_{2j+1}^\dagger \big(1 + Z_{2j-2} X_{2j-1}^\dagger Z_{2j}^{-2} X_{2j+1} Z_{2j+2}\big)$$
$$+ \mathrm{h.c.}. \qquad (2.36)$$

This Hamiltonian indeed preserves the $\{C^o, C^e, K_c\}$ symmetries associated with the cluster model $H_c$. The unique ground state of $H_{c'}$ is fixed by

$$Z_{2j}^\dagger X_{2j+1} Z_{2j+2} = \omega^\alpha, \qquad Y_{2j-1} X_{2j} Y_{2j+1}^\dagger = 1. \qquad (2.37)$$

The Hamiltonian $H_{c'}$ can be mapped to a simpler one through two mutually commuting unitary rotations: the first one is $U_c$ in (2.10) and the second one is

$$W_d = \prod_j \mathrm{CZ}_{2j-2,2j} \mathrm{CZ}_{2j,2j}^\dagger. \qquad (2.38)$$

The subscript $d$ in $W_d$ refers to *dipolar*, as $W_d$ transforms the trivial Hamiltonian $-\sum_j X_{2j}$ to the dipolar cluster model [2] defined on the even sublattice [57]:

$$-\sum_j X_{2j} + \mathrm{h.c.} \xrightarrow{W_d} -\sum_j Z_{2j-2} Z_{2j}^\dagger X_{2j} Z_{2j}^\dagger Z_{2j+2} + \mathrm{h.c.} \qquad (2.39)$$

Under the combined operation $U_{cd}^\dagger = W_d^\dagger U_c^\dagger$,

$$Z_{2j}^\dagger X_{2j+1} Z_{2j+2} \xrightarrow{U_{cd}^\dagger} X_{2j+1},$$
$$Y_{2j-1} X_{2j} Y_{2j+1}^\dagger \xrightarrow{U_{cd}^\dagger} X_{2j-1} X_{2j} X_{2j+1}^\dagger. \qquad (2.40)$$

The SPT Hamiltonian $H_{c'}$ itself transforms to

$$H_{c'} \xrightarrow{U_{cd}^\dagger} -\sum_j \omega^{-\alpha} X_{2j+1} - \sum_j X_{2j-1} X_{2j} X_{2j+1}^\dagger$$
$$- \sum_j X_{2j} + \mathrm{h.c.} \qquad (2.41)$$

The transformed Hamiltonian is written entirely in terms of $X$ operators and its ground state is easy to write down as $g_{2j-1}^X = \alpha$ and $g_{2j}^X = 0$, where $g^X$ is the quantum number in the $X$-basis: $X|g^X\rangle = \omega^{g^X}|g^X\rangle$. The ground state of $H_{c'}$ in (2.36) is $|\psi_{c'}\rangle = W_d U_c |(g^X)^o = \alpha, (g^X)^e = 0\rangle$. By comparison, the cluster ground state is $|\psi_c\rangle = U_c |(g^X)^o = (g^X)^e = 0\rangle$. In the decorated domain wall (DDW) interpretation, the new SPT state is obtained by applying two layers of DDW operations, with each layer consisting of charge domain walls ($U_c$) and dipolar domain wall ($W_d$), respectively.

### 2. Other cSPT states : $H_{c''}$

Using (2.29), we can show that

$$Y_{2j-2} Y_{2j}^\dagger \xrightarrow{V_c} Z_{2j-3} Y_{2j-2} Z_{2j-1}^{-2} Y_{2j}^\dagger Z_{2j+1}, \qquad (2.42)$$

and construct yet another model sharing the symmetries generated by $\{C^o, C^e, V_c\}$:

$$\hat{H}_{c''} = -\sum_j \omega^\alpha Z_{2j-1} Z_{2j+1}^\dagger$$
$$- \sum_j Y_{2j} Y_{2j+2}^\dagger \big(1 + Z_{2j-3} Z_{2j-1}^{-2} Z_{2j+1}\big) + \mathrm{h.c.}. \qquad (2.43)$$

Note that $\hat{H}_{c''}$ represents a family of models parameterized by $\alpha \in \mathbb{Z}_N$. When $N = 2$, the Hamiltonian $\hat{H}_{c''}$ is related to $\hat{H}_{c'}$ via translation $T$ by one site: $\hat{H}_{c''} = T\hat{H}_{c'}T^{-1}$. The ground state conditions for $H_{c''}$ are

$$Z_{2j-1} Z_{2j+1}^\dagger = \omega^{-\alpha}, \qquad Y_{2j} Y_{2j+2}^\dagger = 1. \qquad (2.44)$$

This means $g_{2j-1} = g^o + \alpha j$ for some $g^o \in \mathbb{Z}_N$ and the order parameter for the odd-site state is $\sum_j \omega^{-\alpha j} Z_{2j-1}$. Similarly, $g_{2j}^Y = (g^Y)^e$ and the order parameter for the even-site state is $\sum_j Y_{2j}$. The overall SSB ground states are labeled by two quantum number as $|g^o, (g^Y)^e\rangle$. Compared to the ground states of $\hat{H}_{c'}$, the roles of $g^Y$ and $g$ are interchanged in this case.

Since

$$\omega^{\eta_2} Z_{2j-1} Y_{2j} Z_{2j+1}^\dagger \xrightarrow{(C^o)^{\eta_1}(C^e)^{\eta_2}V_c} Y_{2j}$$
$$\omega^{\eta_1} Z_{2j-1}^\dagger \xrightarrow{(C^o)^{\eta_1}(C^e)^{\eta_2}V_c} Z_{2j-1}, \qquad (2.45)$$

we conclude

$$(C^o)^{\eta_1}(C^e)^{\eta_2}V_c^{\eta_3}|g^o, (g^Y)^e\rangle$$
$$= |\eta_1 + (-1)^{\eta_3}g^o, \eta_2 - \delta_{\eta_3,1}\alpha + (g^Y)^e\rangle. \qquad (2.46)$$

When $\eta_3 = 0$, the only operation that leaves the ground state invariant is $\eta_1 = \eta_2 = 0$, which is an identity. For $\eta_3 = 1$, choosing

$$\eta_1 = 2g^o \ (\mathrm{mod}\ N), \qquad \eta_2 = \alpha \ (\mathrm{mod}\ N), \qquad (2.47)$$

will leave the ground state invariant, with the corresponding invariant symmetry operator given by $(C^o)^{\eta_1}(C^e)^{\eta_2}V_c$. This operator becomes an identity when raised to a power $2\gamma/\gcd(\gamma,2)$, where $\gamma = N/\gcd(N,\eta_2)$. More importantly, the symmetry operation involves some powers of both $C^o$ and $C^e$, in contrast to $\hat{H}_c$ and $\hat{H}_{c'}$ where the invariant symmetry operation $(C^o)^\eta V_c$ involves only $C^o$. When $N = 2$, for $\eta_2 = 0$, the invariant element is just $V_c$, but for $\eta_2 = 1$, a new unbroken symmetry element $C^e V_c$ exists, regardless of the ground state on which it acts.

By applying $\mathrm{KT}_c^\dagger$ on $\hat{H}_{c''}$, one arrives at yet another cSPT model $H_{c''}$, also parameterized by $\alpha$:

$$H_{c''} = -\sum_j \omega^{-\alpha} Z_{2j-1} X_{2j} Z_{2j+1}^\dagger$$
$$- \sum_j Y_{2j}^\dagger X_{2j+1} Y_{2j+2} \big(1 + Z_{2j-1}^\dagger X_{2j}^\dagger Z_{2j+1}^2 X_{2j+2} Z_{2j+3}^\dagger\big)$$
$$+ \mathrm{h.c.}. \qquad (2.48)$$

This Hamiltonian also preserves the $\{C^o, C^e, K_c\}$ symmetries in common with $H_c$ and $H_{c'}$. The unique ground state of $H_{c''}$ is fixed by

$$Z_{2j-1}X_{2j}Z_{2j+1}^\dagger = \omega^\alpha, \qquad Y_{2j}^\dagger X_{2j+1}Y_{2j+2} = 1. \qquad (2.49)$$

The Hamiltonian $H_{c''}$ can be mapped to a simpler one through two mutually commuting unitary rotations: the first one is $U_c$ in (2.10) and the second one is

$$W_{d'} = \prod_j \mathrm{CZ}_{2j-1,2j+1}\mathrm{CZ}_{2j-1,2j-1}^\dagger. \qquad (2.50)$$

$W_d'$ transforms the trivial Hamiltonian $-\sum_j X_{2j-1}$ to the dipolar cluster model [2] defined on the odd sublattice:

$$-\sum_j X_{2j-1} + \text{h.c.} \xrightarrow{W_{d'}} -\sum_j Z_{2j-3}Z_{2j-1}^\dagger X_{2j-1}Z_{2j-1}^\dagger Z_{2j+1} + \text{h.c.} \qquad (2.51)$$

Under the combined operation $U_{cd'}^\dagger = W_{d'}^\dagger U_c^\dagger$,

$$Z_{2j-1}X_{2j}Z_{2j+1}^\dagger \xrightarrow{U_{cd'}^\dagger} X_{2j},$$
$$Y_{2j}^\dagger X_{2j+1}Y_{2j+2} \xrightarrow{U_{cd'}^\dagger} X_{2j}^\dagger X_{2j+1}X_{2j+2}. \qquad (2.52)$$

The SPT Hamiltonian $H_{c''}$ itself transforms to

$$H_{c'} \xrightarrow{U_{cd}^\dagger} -\sum_j \omega^{-\alpha}X_{2j} - \sum_j X_{2j}^\dagger X_{2j+1}X_{2j+2}$$
$$-\sum_j X_{2j}^{-2}X_{2j+1}X_{2j+2}^2 + \text{h.c.} \qquad (2.53)$$

The ground state of the transformed Hamiltonian is given by $g_{2j}^X = \alpha$ and $g_{2j-1}^X = 0$. The ground state of $H_{c''}$ in (2.48) is given by $|\psi_{c''}\rangle = W_{d'}U_c|(g^X)^o = \alpha, (g^X)^e = 0\rangle$, with two layers of DDW. For $N = 2$, $H_{c'}$ and $H_{c''}$ correspond to $H_{\text{even}}$ and $H_{\text{odd}}$ SPT Hamiltonians in [22].

### E. Edge modes

We demonstrate that all three SPT models $H_c, H_{c'}$, and $H_{c''}$ represent distinct phases by interfacing two of the models and examining the symmetry fractionalization taking place at the edges.

To demonstrate the symmetry fractionalization taking place between $H_c$ and $H_{c'}$, we examine the Hamiltonian:

$$H_{c|c'} = -\sum_{j=1}^{l/2} Z_{2j-2}^\dagger X_{2j-1}Z_{2j} - \sum_{j=1}^{l/2-1} Z_{2j-1}X_{2j}Z_{2j+1}^\dagger$$
$$-\sum_{j=l/2+1}^{L/2} \omega^{-\alpha}Z_{2j-1}^\dagger X_{2j-1}Z_{2j}$$
$$-\sum_{j=l/2+1}^{L/2-1} Y_{2j-1}X_{2j}Y_{2j+1}^\dagger\left(1 + Z_{2j-2}X_{2j-1}^\dagger Z_{2j}^{-2}X_{2j+1}Z_{2j+2}\right)$$
$$+ \text{h.c.}. \qquad (2.54)$$

This is a model where $H_c$ occupies $1 \le j \le l$ ($l=$even) sites and $H_{c'}$ occupies $l+1 \le j \le L$ sites, with $L-l$ chosen as a multiple of $2N$. Though the model $H_{c|c'}$ is defined on a periodic chain, there can be multiple ground states which satisfy

$$Z_{2j-2}^\dagger X_{2j-1}Z_{2j} = 1 \qquad (1 \le j \le l/2)$$
$$Z_{2j-1}X_{2j}Z_{2j+1}^\dagger = 1 \qquad (1 \le j \le l/2-1)$$
$$Z_{2j-2}^\dagger X_{2j-1}Z_{2j} = \omega^\alpha \qquad (l/2+1 \le j \le L/2)$$
$$Y_{2j-1}X_{2j}Y_{2j+1}^\dagger = 1 \qquad (l/2+1 \le j \le L/2-1). \qquad (2.55)$$

From these conditions we can infer the action of $C^o$ on the ground state $|\psi\rangle$ as

$$C^o|\psi\rangle = \prod_{j=1}^{L/2} X_{2j-1}|\psi\rangle$$
$$= \prod_{j=1}^{L/2}(Z_{2j-2}^\dagger X_{2j-1}Z_{2j})|\psi\rangle$$
$$= \omega^{\alpha(L-l)/2}|\psi\rangle = |\psi\rangle, \qquad (2.56)$$

the last line following from $L-l$ being a multiple of $2N$. The action of $C^e$ on the ground state, on the other hand, is

$$C^e|\psi\rangle = \prod_{j=1}^{L/2} X_{2j}|\psi\rangle$$
$$= \left(\prod_{j=1}^{l/2-1} Z_{2j-1}X_{2j}Z_{2j+1}^\dagger\right)\tilde{X}_l$$
$$\times \left(\prod_{j=l/2+1}^{L/2-1} Y_{2j-1}X_{2j}Y_{2j+1}^\dagger\right)\tilde{X}_L|\psi\rangle$$
$$= \tilde{X}_l\tilde{X}_L|\psi\rangle, \qquad (2.57)$$

where two fractionalized edge operators emerge:

$$\tilde{X}_l = Z_{l-1}X_lY_{l+1}^\dagger \qquad \tilde{X}_L = Y_{L-1}X_LZ_1^\dagger. \qquad (2.58)$$

With two additional edge operators $Z_l$ and $Z_L$ that commute with the Hamiltonian $H_{c|c'}$, the two pairs of edge operators $(\tilde{X}_l, Z_l)$ and $(\tilde{X}_L, Z_L)$ span the algebra

$$Z_l\tilde{X}_l = \omega\tilde{X}_l\tilde{Z}_l, \qquad Z_L\tilde{X}_L = \omega\tilde{X}_L\tilde{Z}_L, \qquad (2.59)$$

implying $N^2$-fold degenerate ground states.

The fractionalization of $K_c$ can be deduced by examining how $K_c$ acts on the fractionalized operators in (2.58) within the ground state subspace [22]:

$$\tilde{X}_lK_c|\psi\rangle = \omega^{-\alpha}K_c\tilde{X}_l^\dagger|\psi\rangle$$
$$\tilde{X}_LK_c|\psi\rangle = \omega^\alpha K_c\tilde{X}_L^\dagger|\psi\rangle$$
$$Z_l^\dagger Z_LK_c|\psi\rangle = K_cZ_lZ_L^\dagger|\psi\rangle$$
$$K_c^2|\psi\rangle = N\sum_{k=1}^N (\tilde{X}_l\tilde{X}_L)^k|\psi\rangle. \qquad (2.60)$$

An explicit expression for $K_c$ that satisfies these relations is given by:

$$K_c|\psi\rangle = \varepsilon P Z_l^{-\alpha} Z_L^{\alpha} \left( \sum_{k=1}^{N} (\tilde{X}_L \tilde{X}_R)^k \right) |\psi\rangle$$

$$= \varepsilon P \left( \sum_{k=1}^{N} K_{c,l}^{(k)} K_{c,L}^{(k)} \right) |\psi\rangle, \qquad (2.61)$$

where we introduced the abbreviations

$$K_{c,l}^{(k)} = Z_l^{-\alpha} \tilde{X}_l^k, \qquad K_{c,L}^{(k)} = Z_L^{\alpha} \tilde{X}_L^k.$$

The prefactor $\varepsilon$ is either +1 or -1, but its exact value does not affect the ensuing discussion.

Setting $K_c \to P\left(\sum_{k=1}^{N} K_{c,l}^{(k)} K_{c,L}^{(k)}\right)$ without loss of generality, we arrive at the algebra among the fractionalized symmetry operators:

$$K_{c,l}^{(k)} \tilde{X}_l = \omega^{-\alpha} \tilde{X}_l K_{c,l}^{(k)}, \qquad K_{c,L}^{(k)} \tilde{X}_L = \omega^{\alpha} \tilde{X}_L K_{c,L}^{(k)} \qquad (2.62)$$

and confirm that they form projective representations for all $\alpha \neq 0$. We conclude that $H_{c'}$ with $\alpha = 0$ lies in the same SPT phase as the cluster state but $H_{c'}$ with $\alpha \neq 0$ realizes a new SPT phase though both $H_c$ and $H_{c'}$ are protected by the same set of symmetries. In particular, $C^o$ reduces to an identity while $C^e$ fractionalizes as shown in (2.57). This means that the two phases of $H_c$ and $H_{c'}$ are distinguished only by virtue of the presence of the KW symmetry $K_c$ and its fractionalization with respect to one of the global symmetries, $C^e$.

It is straightforward to write down an interfacial model between $H_c$ and $H_{c''}$ or even between $H_{c'}$ and $H_{c''}$ and go through the same exercise as above to demonstrate symmetry fractionalizations. Besides, we can demonstrate that $H_{c'}$'s with different $\alpha$'s correspond to distinct SPT phases, thus represent a family of SPTs distinct from the cluster state as well as from one another. Calculations are mostly variations on what has been covered in this subsection, and can be found in the Appendix B and C.

We conclude this section with a note that an alternative approach to distinguish the various cSPT phases, besides the symmetry fractionalization at the interface of the two cSPTs, is to compute the strange correlators between two cSPTs [58]. Furthermore, Ref. [59] employs the SymTFT framework to classify distinct cSPT phases, demonstrating that their total number is given by $N^2 f(N)$ for odd $N$ and $\frac{3}{4}N^2 f(N)$ for even $N$, where $f(N)$ denotes the number of integers $k$ satisfying $\gcd(N,k) = d$ with $d^2 \equiv 0 \pmod{N}$.

## F. cSPT from holography

From the perspective of topological holography, a $d$-dimensional theory with a finite symmetry group $S$ can be viewed as the boundary of a $(d+1)$-dimensional topological quantum field theory, known as a symmetry TFT (SymTFT) [9, 28, 60–62]. In this framework, the symmetry information and the dynamical information are separately encoded on two distinct boundaries. For instance, the $\mathbb{Z}_N$ transverse Ising model, which emerges as the boundary effective theory of the $\mathbb{Z}_N$ toric code [21], serves as a clear example of topological holography: the $\mathbb{Z}_N$ charge symmetry in the Ising model is inherited from the 1-form symmetry of the bulk toric code, and the charged operators in the Ising model correspond to the endpoints of charge strings that extend from the bulk to the boundary.

When considering the boundary theory of two coupled toric codes, one can explore a variety of boundary conditions or anyon condensation schemes. One such choice leads to the 1D cluster state as the (1+1)D effective boundary theory of the bilayer toric code. The $\mathbb{Z}_N \times \mathbb{Z}_N$ symmetry of the cluster state can be traced back to the Wilson loop operators (1-form symmetries), one from each toric code layer. Rather than detailing this construction here, we defer a full discussion to Sec. IV, where we introduce a generalized toric code and analyze its boundary theory. The coupled toric codes and their relation to the cluster state then emerge as a special case of that framework.

## III. DIPOLE SYMMETRY

One way to generalize the charge symmetry is to impose an additional multipolar symmetry such as the dipole symmetry on. Another way is to keep the charge symmetry, but elevate it to an exponentially modulated variety. Both schemes fall in the broad category of modulated symmetries and there are SPT phases protected by them [2]. In this section, we consider the noninvertible symmetry associated with gauging the dipole symmetry and investigate SPT phases protected by it. The case of exponentially modulated charge symmetry will be discussed in Sec. IV.

### A. Gauging and noninvertible dipole symmetry

We consider the gauging of dipole symmetry $D = \prod_j X_j^j$ along with the charge symmetry $C = \prod_j X_j$ on an infinite chain [5, 6]. The local gauge symmetry operator

$$g_j = X_j \bar{Z}_{j-1}^{\dagger} \bar{Z}_j^2 \bar{Z}_{j+1}^{\dagger} \qquad (3.1)$$

involves gauge fields $(\bar{X}_j, \bar{Z}_j)$ and matter fields $(X_j, Z_j)$. The global symmetries are expressed in terms of $g_j$ as

$$C = \prod_j g_j, \qquad D = \prod_j g_j^j. \qquad (3.2)$$

The matter field operators that commute with $C$ and $D$ are $X_j$ and $Z_{j-1} Z_j^{-2} Z_{j+1}$, which transform under the gauging as

$$X_j \to \bar{Z}_{j-1} \bar{Z}_j^{-2} \bar{Z}_{j+1},$$
$$Z_{j-1}^{\dagger} Z_j^2 Z_{j+1}^{\dagger} \to \bar{X}_j. \qquad (3.3)$$

This mapping is implemented by the dipolar Kramers-Wannier (dKW) operator $K_{\text{dKW}}$ [5]:

$$K_{\text{dKW}} = \sum_{\mathbf{g},\mathbf{g}'} \omega^{\sum_j (g_{j+1} - 2g_j + g_{j-1})g'_j} |\mathbf{g}'\rangle\langle\mathbf{g}|. \qquad (3.4)$$

Dipolar SPT (dSPT) phases protected by one charge and one dipole symmetry were proposed in [2]. A simple model explicitly realizing dSPT order is [5]

$$H_d^{(k)} = -\sum_j \sum_{m=1}^N \left( (Z_{j-1}Z_j^\dagger)^k X_j (Z_j^\dagger Z_{j+1})^k \right)^m. \qquad (3.5)$$

This model is given as the sum over all $m$-th powers $1 \le m \le N$, in contrast to [2] where only the $m = 1$ and $m = N-1$ were considered in the definition of the model. While not changing the symmetry of the model, this change in the sum over $m$ has an important bearing on the consideration of noninvertible symmetry.

Implementing $K_{\text{dKW}}$ on $H_d^{(k)}$ gives the dual Hamiltonian

$$H_d^{(k)} \xrightarrow{K_{\text{dKW}}} -\sum_j \sum_m \left( Z_{j-1}Z_j^\dagger X_j^{-k} Z_j^\dagger Z_{j+1} \right)^m, \qquad (3.6)$$

which equals the original model $H_d^{(k)}$ only if $k^2 \equiv -1$ mod $N$ [5]. Since the dipolar SPT model is well-defined only for $N > 2$, the smallest integer values of $(k, N)$ for which the noninvertible symmetry holds is $(k, N) = (2, 5)$. Even in this case, the noninvertible symmetry holds only when we allow a full summation over $m \in \mathbb{Z}_N$.

### B. Dipolar SPT

Although the dipolar cluster model $H_d^{(k)}$ does possess charge and dipole symmetries, *and* the noninvertible symmetry for special choices of $k$, it seems desirable to work with a model for dSPT that can embody the noninvertible symmetry in a more natural way. The following model meets such criterion:

$$H_d = -\sum_j Z_{2j-1} X_{2j} Z_{2j+1}^{-2} Z_{2j+3}$$
$$-\sum_j Z_{2j-2} Z_{2j}^{-2} X_{2j+1} Z_{2j+2} + \text{h.c.}. \qquad (3.7)$$

This model has the charge operator $X$ dressed by quadrupole operator $ZZ^{-2}Z$ (dipole-antidipole domain wall) defined exclusively over the even (odd) sites in the case of odd-site (even-site) $X$. There are *two charge and two dipole* symmetry operators associated with this model:

$$C^o = \prod_j X_{2j-1}, \qquad C^e = \prod_j X_{2j},$$
$$D^o = \prod_j X_{2j-1}^j, \qquad D^e = \prod_j X_{2j}^j. \qquad (3.8)$$

Here, dipole symmetries are modulated in the sense that the translation by two lattice sites results in non-trivial change of the symmetry operator: $T^{-2}D^{o(e)}T^2 = D^{o(e)}C^{o(e)}$.

The ground state is given by

$$|\psi_d\rangle \propto \sum_{\mathbf{g}} \omega^{\sum_j g_{2j}(g_{2j-1} - 2g_{2j+1} + g_{2j+3})} |\mathbf{g}\rangle$$
$$= \sum_{\mathbf{g}} \omega^{\sum_j g_{2j+1}(g_{2j+2} - 2g_{2j} + g_{2j-2})} |\mathbf{g}\rangle, \qquad (3.9)$$

and can be obtained by the CZ operation

$$U_d = \prod_j CZ_{2j,2j-1} CZ_{2j,2j+1}^{-2} CZ_{2j,2j+3}$$
$$= \prod_j CZ_{2j+1,2j-2} CZ_{2j+1,2j+2} CZ_{2j+1,2j}^{-2} \qquad (3.10)$$

on the paramagnetic state $|+\rangle = \prod_j |+\rangle_j$. For proof, note that

$$X_{2j} \xrightarrow{U_d} Z_{2j-1} X_{2j} Z_{2j+1}^{-2} Z_{2j+3},$$
$$X_{2j+1} \xrightarrow{U_d} Z_{2j-2} Z_{2j}^{-2} X_{2j+1} Z_{2j+2}, \qquad (3.11)$$

and thus $-\sum_j (X_j + X_j^\dagger) \xrightarrow{U_d} H_d$.

Crucial to our agenda is the fact that the model exhibits a noninvertible symmetry $K_d H_d = H_d K_d$ with

$$K_d = \left( \prod_j \text{SWAP}_{2j,2j+1} \right) K_{\text{dKW}}^o (K_{\text{dKW}}^e)^\dagger, \qquad (3.12)$$

and $K_{\text{dKW}}^o$ and $K_{\text{dKW}}^e$ are the dKW operators introduced in (3.4) acting on the odd and even sublattices, respectively. $\text{SWAP}_{2j,2j+1}$ is the SWAP gate acting on the qudits at $2j$ and $2j+1$. The $K_d$ performs the transformation

$$X_{2j} \xrightarrow{K_d} Z_{2j-1}^\dagger Z_{2j+1}^2 Z_{2j+3}^\dagger,$$
$$X_{2j+1} \xrightarrow{K_d} Z_{2j-2} Z_{2j}^{-2} Z_{2j+2},$$
$$Z_{2j-2} Z_{2j}^{-2} Z_{2j+2} \xrightarrow{K_d} X_{2j+1},$$
$$Z_{2j-1}^\dagger Z_{2j+1}^2 Z_{2j+3}^\dagger \xrightarrow{K_d} X_{2j}, \qquad (3.13)$$

and preserves $H_d$: $K_d H_d = H_d K_d$. Altogether, there are two charge symmetries, two dipole symmetries, and one noninvertible symmetry associated with $H_d$ which generate the fusion algebra:

$$C^o K_d = K_d C^o = K_d, \qquad C^e K_d = K_d C^e = K_d,$$
$$D^o K_d = K_d D^o = K_d, \qquad D^e K_d = K_d D^e = K_d,$$
$$K_d^\dagger K_d = \left( \sum_k (C^e)^k \right) \left( \sum_k (C^o)^k \right) \left( \sum_k (D^e)^k \right) \left( \sum_k (D^o)^k \right). \qquad (3.14)$$

The unitarity of $K_d$ is recovered in the symmetric sector $C^o = C^e = D^o = D^e = 1$.

### C. Kennedy-Tasaki transformation

The KT transformation for the dipolar cluster model is implemented by [5]

$$\text{KT}_d = U_d K_d U_d^\dagger$$

where $U_d$ defined in (3.10) maps the paramagnetic state $\sum_{\mathbf{g}} |\mathbf{g}\rangle$ to the cluster ground state. One can show

$$X_{2j} \xrightarrow{\text{KT}_d} X_{2j},$$

$$X_{2j+1} \xrightarrow{\text{KT}_d} X_{2j+1}^\dagger,$$

$$Z_{2j-2} Z_{2j}^{-2} Z_{2j+2} \xrightarrow{\text{KT}_d} Z_{2j-2} Z_{2j}^{-2} X_{2j+1} Z_{2j+2},$$

$$Z_{2j-1}^\dagger Z_{2j+1}^2 Z_{2j+3}^\dagger \xrightarrow{\text{KT}_d} Z_{2j-1} X_{2j} Z_{2j+1}^{-2} Z_{2j+3}. \quad (3.15)$$

The symmetry operators of the cluster model transform as

$$C^o \xrightarrow{\text{KT}_d} (C^o)^\dagger, \qquad C^e \xrightarrow{\text{KT}_d} C^e,$$

$$D^o \xrightarrow{\text{KT}_d} (D^o)^\dagger, \qquad D^e \xrightarrow{\text{KT}_d} D^e. \quad (3.16)$$

The noninvertible symmetry $K_d$ under the conjugation by $\text{KT}_d$ becomes

$$\text{KT}_d \cdot K_d = \left(U_d P^o\right) \text{KT}_d \equiv V_d \cdot \text{KT}_d. \quad (3.17)$$

In summary, the symmetries of the dipolar cluster model transform to a new set of symmetries

$$\{C^o, C^e, D^o, D^e, K_d\} \xrightarrow{\text{KT}_d} \{(C^o)^\dagger, C^e, (D^o)^\dagger, D^e, V_d\}, \quad (3.18)$$

all of which are invertible and unitary. Since $V_d^2 = 1$, the symmetry group generated by $V_d$ is $\mathbb{Z}_2$. For general $N$, $V_d$ has a non-trivial commutation relation with $C^o$, $C^e$, $D^o$, and $D^e$, and the overall symmetry group is described as

$$\mathbb{Z}_N^{c,e} \times \mathbb{Z}_N^{d,e} \times \left[\left(\mathbb{Z}_N^{c,o} \times \mathbb{Z}_N^{d,o}\right) \rtimes \mathbb{Z}_2^{V_d}\right] \quad (3.19)$$

with $\mathbb{Z}_2^{V_d}$ acting as an outer automorphism on $\mathbb{Z}_N^{c,o} \times \mathbb{Z}_N^{d,o}$.

Conjugating the dipolar cluster model $H_d$ by $\text{KT}_d$ results in two copies of $\mathbb{Z}_N$ dipolar Ising model, or double dipolar Ising model (dIM2),

$$\hat{H}_d = -\sum_j Z_{j-2} Z_j^{-2} Z_{j+2} + \text{h.c.}. \quad (3.20)$$

One can explicitly check that dIM2 possesses all the symmetries shown on the right side of (3.19). The ground states are

characterized by $g_{j+2} - g_j = g_j - g_{j-2}$ on the even and odd sublattice separately. There are altogether four quantum numbers $g_c^o, g_c^e, g_d^o, g_d^e \in \mathbb{Z}_N$ to characterize the ground state:

$$g_{2j} = g_c^e + g_d^e \cdot j, \qquad g_{2j+1} = g_c^o + g_d^o \cdot j \pmod{N}. \quad (3.21)$$

The order parameters are $\sum_j Z_{2j}^\dagger Z_{2j+2}$, $\sum_j Z_{2j-1}^\dagger Z_{2j+1}$, $\sum_j Z_{2Nj}$, and $\sum_j Z_{2Nj+1}$.

The ground state $|g_c^o, g_c^e, g_d^o, g_d^e\rangle$ breaks all the symmetries of the dIM2 model, yet the GSD is only $N^4$ not $2N^4$. The action by an element of the symmetry group on the ground state yields

$$(C^o)^{\eta_1} (C^e)^{\eta_2} (D^o)^{\eta_3} (D^e)^{\eta_4} V_d^{\eta_5} |g_c^o, g_c^e, g_d^o, g_d^e\rangle$$
$$= |\eta_1 + (-1)^{\eta_5} g_c^o, \eta_2 + g_c^e, \eta_3 + (-1)^{\eta_5} g_d^o, \eta_4 + g_d^e\rangle, \quad (3.22)$$

where $\eta_1, \eta_2, \eta_3, \eta_4 \in \mathbb{Z}_N$ and $\eta_5 \in \mathbb{Z}_2$. These states collectively span a Hilbert space of dimension $N^4$.

When $\eta_5 = 1$, the state $|g_c^o, g_c^e, g_d^o, g_d^e\rangle$ remains invariant under the action of $(C^o)^{\eta_1} (C^e)^{\eta_2} (D^o)^{\eta_3} (D^e)^{\eta_4} V_d$ provided that other parameters satisfy the condition

$$\eta_1 = 2g_c^o \pmod{N}, \qquad \eta_2 = 0$$
$$\eta_3 = 2g_d^o \pmod{N}, \qquad \eta_4 = 0. \quad (3.23)$$

The invariant symmetry operator $(C^o)^{\eta_1} (D^o)^{\eta_3} V_d$ squares to one.

## D. Other dSPT states

One-dimensional dSPT phases protected by $C$ and $D$ symmetries in (3.2) are classified by the cohomology $H^2(\mathbb{Z}_N^2, U(1))/[H^2(\mathbb{Z}_N, U(1))]^2 = \mathbb{Z}_N$ [3, 7]. A similar reasoning shows that 1D SPT phases protected by $\{C^o, C^e, D^o, D^e\}$ symmetries of (3.8) are classified by

$$H^2(\mathbb{Z}_N^4, U(1))/[H^2(\mathbb{Z}_N^2, U(1))]^2 = \mathbb{Z}_N^4. \quad (3.24)$$

One can write down explicit models realizing each of these SPTs:

$$H_d^{(k_1, k_2, k_3, k_4)} = -\sum_j X_{2j} \cdot (Z_{2j-1} Z_{2j+1}^{-2} Z_{2j+3})^{k_1} (\omega^{-1} Z_{2j-2} Z_{2j}^{-2} Z_{2j+2})^{k_2} (Z_{2j-3} Z_{2j-1}^{-3} Z_{2j+1}^3 Z_{2j+3}^\dagger)^{k_4}$$
$$- \sum_j X_{2j+1} \cdot (Z_{2j-2} Z_{2j}^{-2} Z_{2j+2})^{k_1} (\omega^{-1} Z_{2j-1} Z_{2j+1}^{-2} Z_{2j+3})^{k_3} (Z_{2j-2}^\dagger Z_{2j}^3 Z_{2j+2}^{-3} Z_{2j+4})^{k_4} + \text{h.c.}, \quad (3.25)$$

where $k_1, k_2, k_3, k_4 \in \mathbb{Z}_N$. The Hamiltonian in (3.7) corresponds to a special case with $k_1 = 1$, $k_2 = k_3 = k_4 = 0$. Within the family of models $H_d^{(k_1, k_2, k_3, k_4)}$, $H_d^{(1,0,0,0)}$, $H_d^{(0,1,-1,0)}$, and $H_d^{(0,-1,1,0)}$ preserve the symmetry $K_d$ under PBC. Performing $K_d$ on $H_d^{(0,1,-1,0)}$ or $H_d^{(0,-1,1,0)}$ results in a unit translation of

the terms in the Hamiltonian, while no such translation takes place for $H_d^{(1,0,0,0)}$. This has a subtle but important effect when the model is placed on a finite segment of the chain such as in the consideration of the fractionalized edge modes at the interface with another SPT. For this reason, we will focus on

the analysis of the $H_d^{(1,0,0,0)} = H_d$ model in the remainder of the paper. The way we arrived at (3.25) and other details can be found in the Appendix.

We now construct an alternative Hamiltonian that preserves the same symmetries $\{C^o, C^e, D^o, D^e, V_d\}$ as dIM2. As before, we introduce $Y \equiv \omega^{1/2} XZ$ which transforms under $V_d$ as

$$Y_{2j} \xrightarrow{V_d} Z_{2j-1} Y_{2j} Z_{2j+1}^{-2} Z_{2j+3},$$
$$Y_{2j+1} \xrightarrow{V_d} Z_{2j-2}^\dagger Z_{2j}^2 Y_{2j+1}^\dagger Z_{2j+2}^\dagger,$$
$$Z_{2j} \xrightarrow{V_d} Z_{2j},$$
$$Z_{2j+1} \xrightarrow{V_d} Z_{2j+1}^\dagger. \tag{3.26}$$

Since

$$Y_{2j-1} Y_{2j+1}^{-2} Y_{2j+3} \xrightarrow{V_d}$$
$$Z_{2j-4}^\dagger Z_{2j-2}^4 Y_{2j-1}^\dagger Z_{2j}^{-6} Y_{2j+1}^2 Z_{2j+2}^4 Y_{2j+3}^\dagger Z_{2j+4}^\dagger, \tag{3.27}$$

the following model satisfies all the symmetries shown on the right side of (3.18):

$$\hat{H}_{d'} = -\sum_j \omega^{\alpha+\beta j} Z_{2j-2} Z_{2j}^{-2} Z_{2j+2}$$
$$- \sum_j Y_{2j-1} Y_{2j+1}^{-2} Y_{2j+3} (1 + Z_{2j-4} Z_{2j-2}^{-4} Z_{2j}^6 Z_{2j+2}^{-4} Z_{2j+4})$$
$$+ \text{h.c.}. \tag{3.28}$$

A pair of $\mathbb{Z}_N$ integers $(\alpha, \beta)$ parametrizes the model. The first term in $\hat{H}_{d'}$ is minimized by $Z_{2j-2} Z_{2j}^{-2} Z_{2j+2} = \omega^{-\alpha-\beta j}$. The

product of $Z$'s in the second term can be represented as

$$Z_{2j-4} Z_{2j-2}^{-4} Z_{2j}^6 Z_{2j+2}^{-4} Z_{2j+4}$$
$$= Z_{2j-4} Z_{2j-2}^{-2} Z_{2j} \cdot (Z_{2j-2} Z_{2j}^{-2} Z_{2j+2})^{-2} \cdot Z_{2j} Z_{2j+2}^{-2} Z_{2j+4}, \tag{3.29}$$

and equals one in the ground state. The overall ground state conditions are

$$Z_{2j-2} Z_{2j}^{-2} Z_{2j+2} = \omega^{-\alpha-\beta j}, \quad Y_{2j-1} Y_{2j+1}^{-2} Y_{2j+3} = 1. \tag{3.30}$$

The ground states $|(g_c^Y)^o, (g_c)^e, (g_d^Y)^o, g_d^e\rangle$ can be labeled by four $\mathbb{Z}_N$ integers satisfying

$$g_{2j} = g_c^e + g_d^e j - \alpha \frac{(j-1)j}{2} - \beta \frac{(j-1)j(j+1)}{6},$$
$$g_{2j-1}^Y = (g_c^Y)^o + (g_d^Y)^o \cdot j. \tag{3.31}$$

The order parameters are $\sum_j g_{2Nj}$, $\sum_j g_{2j-2}^\dagger g_{2j} \omega^{\alpha j + \beta j(j+1)/2}$, $\sum_j g_{2Nj-1}^Y$, $\sum_j (g_{2j-1}^Y)^\dagger g_{2j+1}^Y$. As in the dIM2, there are $N^4$ ground states although all five symmetries of the model $\hat{H}_{d'}$ are broken.

Since

$$\omega^{\eta_1 + \eta_3 j} Z_{2j-4}^\dagger Z_{2j-2}^4 Y_{2j-1}^\dagger Z_{2j}^\dagger \xrightarrow{(C^o)^{\eta_1}(C^e)^{\eta_2}(D^o)^{\eta_3}(D^e)^{\eta_4} V_d} Y_{2j-1}$$
$$\omega^{\eta_2 + \eta_4 j} Z_{2j} \xrightarrow{(C^o)^{\eta_1}(C^e)^{\eta_2}(D^o)^{\eta_3}(D^e)^{\eta_4} V_d} Z_{2j}, \tag{3.32}$$

the ground state quantum numbers undergo the transformation

$$(C^o)^{\eta_1}(C^e)^{\eta_2}(D^o)^{\eta_3}(D^e)^{\eta_4} V_d^{\eta_5} |(g_c^Y)^o, (g_c)^e, (g_d^Y)^o, g_d^e\rangle$$
$$= |\eta_1 + \delta_{\eta_5,1}\alpha + (-1)^{\eta_5}(g_c^Y)^o, \eta_2 + g_c^e, \eta_3 + \delta_{\eta_5,1}\beta + (-1)^{\eta_5}(g_d^Y)^o, \eta_4 + g_d^e\rangle,$$

under the action by the symmetry group elements. The condition to leave the ground state invariant is

$$\eta_1 = 2(g_c^Y)^o - \alpha \pmod{N}, \quad \eta_2 = 0,$$
$$\eta_3 = 2(g_d^Y)^o - \beta \pmod{N}, \quad \eta_4 = 0, \quad \eta_5 = 1. \tag{3.33}$$

The invariant symmetry operator $(C^o)^{\eta_1}(D^o)^{\eta_3} V_d$ is $\mathbb{Z}_2$

squares to one. This looks identical to the invariant symmetry element of the dIM2, but the structure of the ground states as well as the rules for fixing $(\eta_1, \eta_3)$ are different in the two models - see (3.23) and (3.33).

Applying $\text{KT}_d^\dagger$ on the Hamiltonian (3.28) gives $\hat{H}_{d'} \xrightarrow{\text{KT}_d^\dagger} H_{d'}$ where

$$H_{d'} = -\sum_j \omega^{\alpha+\beta j} Z_{2j-2} Z_{2j}^{-2} X_{2j+1} Z_{2j+2} - \sum_j Y_{2j-1} X_{2j} Y_{2j+1}^{-2} Y_{2j+3} \left(1 + Z_{2j-4}^\dagger Z_{2j-2}^4 X_{2j-1}^\dagger Z_{2j}^{-6} X_{2j+1}^2 Z_{2j+2}^4 X_{2j+3}^\dagger Z_{2j+4}^\dagger\right) + \text{h.c.} \tag{3.34}$$

The same $\{C^o, C^e, D^o, D^e, K_d\}$ symmetries exist for both the original dipolar cluster model $H_d$ and this one. The unique

ground state of $H_{d'}$ is fixed by

$$Z_{2j-2} Z_{2j}^{-2} X_{2j+1} Z_{2j+2} = \omega^{-\alpha-\beta j},$$
$$Y_{2j-1} X_{2j} Y_{2j+1}^{-2} Y_{2j+3} = 1. \tag{3.35}$$

The lengthy expression inside the parenthesis of (3.34) equals 1 by virtue of the first condition.

The new dSPT Hamiltonian $H_{d'}$ in (3.34) can be brought to a simpler form through two successive unitary rotations: $U_d$ in (3.10) and

$$W_o = \prod_j CZ_{2j-4,2j}^\dagger CZ_{2j-2,2j}^4 CZ_{2j,2j}^{-3}. \qquad (3.36)$$

The subscript $o$ refers to *octupolar*, as $W_o$ transforms the trivial Hamiltonian $-\sum_j X_{2j}$ to the octupolar cluster model on the even sublattice:

$$-\sum_j X_{2j} + \text{h.c.} \xrightarrow{W_o} -\sum_j Z_{2j-4}^\dagger Z_{2j-2}^4 Z_{2j}^{-3} X_{2j} Z_{2j}^{-3} Z_{2j+2}^4 Z_{2j+4}^\dagger$$
$$+ \text{h.c.} \qquad (3.37)$$

Under the combined operation $U_{do}^\dagger = W_o^\dagger U_d^\dagger$,

$$Z_{2j-2} Z_{2j}^{-2} X_{2j+1} Z_{2j+2} \xrightarrow{U_{do}^\dagger} X_{2j+1},$$

$$Y_{2j-1} X_{2j} Y_{2j+1}^{-2} Y_{2j+3} \xrightarrow{U_{do}^\dagger} X_{2j-1} X_{2j} X_{2j+1}^{-2} X_{2j+3}. \qquad (3.38)$$

The Hamiltonian itself transforms to

$$H_{d'} \xrightarrow{U_{do}^\dagger} -\sum_j \omega^{\alpha+\beta j} X_{2j+1} - \sum_j X_{2j-1} X_{2j} X_{2j+1}^{-2} X_{2j+3}$$
$$- \sum_j X_{2j} + \text{h.c.} \qquad (3.39)$$

The ground state of this Hamiltonian is given by

$$g_{2j+1}^X = -\alpha - \beta j, \quad g_{2j}^X = 0.$$

The ground state of the new SPT Hamiltonian $H_{d'}$ itself is given by applying $U_{do} = W_o U_d$ on it. Each layer of the unitaries can be interpreted as the decoration by dipolar domain walls ($U_d$), followed by the second layer of octupolar domain walls decorating the even sites ($W_o$).

One may attempt to construct other Hamiltonians sharing the same symmetry group as DIM2 but whose ground states are invariant under a different set of symmetry elements. We feel that such analysis can be done but may not result in particularly new insights.

### E. Edge modes

We reveal the distinct nature of the two dipolar SPT Hamiltonians $H_d$ and $H_{d'}$ by examining the symmetry fractionalization in the Hamiltonian:

$$H_{d|d'} = -\sum_{j=1}^{l/2} Z_{2j-4} Z_{2j-2}^{-2} X_{2j-1} Z_{2j} - \sum_{j=0}^{l/2-2} Z_{2j-1} X_{2j} Z_{2j+1}^{-2} Z_{2j+3} - \sum_{j=l/2+1}^{L/2} \omega^{\alpha+\beta j} Z_{2j-4} Z_{2j-2}^{-2} X_{2j-1} Z_{2j}$$
$$- \sum_{j=l/2+1}^{L/2-2} Y_{2j-1} X_{2j} Y_{2j+1}^{-2} Y_{2j+3} \left( 1 + Z_{2j-4}^\dagger Z_{2j-2}^4 X_{2j-1}^\dagger Z_{2j}^{-6} X_{2j+1}^2 Z_{2j+2}^4 X_{2j+3}^\dagger Z_{2j+4}^\dagger \right) + \text{h.c.} \qquad (3.40)$$

where $L$ is a multiple of $2N$, $L - l$ is a multiple of $12N$. This Hamiltonian consists of $H_d$ over $1 \le j \le l$ and $H_d'$ defined over the remaining sites $l + 1 \le j \le L$. The ground states should satisfy the conditions:

$$Z_{2j-4} Z_{2j-2}^{-2} X_{2j-1} Z_{2j} = 1 \qquad (1 \le j \le l/2)$$
$$Z_{2j-1} X_{2j} Z_{2j+1}^{-2} Z_{2j+3} = 1 \qquad (0 \le j \le l/2 - 2)$$
$$Z_{2j-4} Z_{2j-2}^{-2} X_{2j-1} Z_{2j} = \omega^{-\alpha-\beta j} \quad (l/2 + 1 \le j \le L/2)$$
$$Y_{2j-1} X_{2j} Y_{2j+1}^{-2} Y_{2j+3} = 1 \qquad (l/2 + 1 \le j \le L/2 - 2)$$
$$(3.41)$$

The action of symmetry operators $C^o$, $C^e$, $D^o$, and $D^e$ within the ground state subspace can be expressed as:

$$C^o |\psi\rangle = |\psi\rangle, \qquad C^e |\psi\rangle = \tilde{X}_{l,l+2} \tilde{X}_{L,2} |\psi\rangle,$$
$$D^o |\psi\rangle = |\psi\rangle, \qquad D^e |\psi\rangle = \check{X}_{l+2} \check{X}_2 |\psi\rangle, \qquad (3.42)$$

where $|\psi\rangle$ is a ground state of $H_{d|d'}$. Various fractionalized operators are deduced by examining the action of global symmetry operators on the ground states:

$$\tilde{X}_{l-2,l} = Z_{l-3} X_{l-2} Z_{l-1}^\dagger X_l Y_{l+1}^\dagger Y_{l+3}$$
$$\tilde{X}_{L-2,L} = Y_{L-3} X_{L-2} Y_{L-1}^\dagger X_L Z_1^\dagger Z_3$$
$$\check{X}_l = Z_{l-1} X_l Y_{l+1}^{-2} Y_{l+3}$$
$$\check{X}_L = Y_{L-1} X_L Z_1^{-2} Z_3. \qquad (3.43)$$

Subscripts underlying the operators refer to the location of the $X$ operators. The fractionalized charge operators (denoted by $\tilde{X}$) carry two spatial indices, while the fractionalized dipole operators (denoted by $\check{X}$) carry only one spatial index.

Additionally, we have four operators $Z_{l-2}$, $Z_l$, $Z_{L-2}$, and $Z_L$ which commute with the Hamiltonian $H_{d|d'}$. These are defined at the sites where the stabilizers are "missing". The commu-

tation relations among the fractionalized operators are given by:

$$Z_{l-2}\tilde{X}_{l-2,l} = \omega\tilde{X}_{l-2,l}Z_{l-2} \qquad Z_{l-2}\check{X}_l = \check{X}_l Z_{l-2}$$
$$Z_l\tilde{X}_{l-2,l} = \omega\tilde{X}_{l-2,l}Z_l \qquad Z_l\check{X}_l = \omega\check{X}_l Z_l$$
$$Z_{L-2}\tilde{X}_{L-2,L} = \omega\tilde{X}_{L-2,L}Z_{L-2} \qquad Z_{L-2}\check{X}_L = \check{X}_L Z_{L-2}$$
$$Z_L\tilde{X}_{L-2,L} = \omega\tilde{X}_{L-2,L}Z_L \qquad Z_L\check{X}_L = \omega\check{X}_L Z_L. \qquad (3.44)$$

The action of $K_d$ on the fractionalized symmetry operators in (3.43) within the ground state subspace is summarized as follows:

$$\tilde{X}_{l-2,l}K_d|\psi\rangle = \omega^{-\beta}K_d\tilde{X}_{l-2,l}^\dagger|\psi\rangle$$
$$\tilde{X}_{L-2,L}K_d|\psi\rangle = \omega^{-\alpha}K_d\tilde{X}_{L-2,L}^\dagger|\psi\rangle$$
$$\check{X}_l K_d|\psi\rangle = \omega^{\alpha+\beta}K_d\check{X}_l^\dagger|\psi\rangle$$
$$\check{X}_L K_d|\psi\rangle = K_d\check{X}_L^\dagger|\psi\rangle$$
$$Z_{l-2}Z_{L-2}^\dagger K_d|\psi\rangle = K_d Z_{l-2}^\dagger Z_{L-2}|\psi\rangle$$
$$Z_l Z_L^\dagger K_d|\psi\rangle = \omega^{-\alpha-2\beta}K_d Z_l^\dagger Z_L|\psi\rangle$$
$$K_d^2|\psi\rangle = N^2\left(\sum_{k_1,k_2=1}^{N}(\tilde{X}_{l-2,l}\tilde{X}_{L-2,L})^{k_1}(\check{X}_{l+2}\check{X}_2)^{k_2}\right)|\psi\rangle.$$
$$(3.45)$$

An explicit expression for the fractionalized dKW operator $K_d$ that satisfies these relations is given by:

$$K_d|\psi\rangle = \varepsilon P Z_{l-2}^{-\alpha-2\beta} Z_l^{\alpha+\beta} Z_{L-2}^{-\alpha}\check{X}_L^{-\alpha-2\beta}$$
$$\times\left(\sum_{k_1,k_2=1}^{N}(\tilde{X}_{l-2,l}\tilde{X}_{L-2,L})^{k_1}(\check{X}_l\check{X}_L)^{k_2}\right)|\psi\rangle$$
$$= \varepsilon P\left(\sum_{k_1,k_2=1}^{N}K_{d,l}^{(k_1,k_2)}K_{d,L}^{(k_1,k_2)}\right)|\psi\rangle, \qquad (3.46)$$

where $\varepsilon$ is either +1 or -1, and

$$K_{d,l}^{(k_1,k_2)} = Z_{l-2}^{-\alpha-2\beta}Z_l^{\alpha+\beta}\tilde{X}_{l-2,l}^{k_1}\check{X}_l^{k_2},$$
$$K_{d,L}^{(k_1,k_2)} = Z_{L-2}^{-\alpha}\tilde{X}_{L-2,L}^{k_1}\check{X}_L^{k_2-\alpha-2\beta}. \qquad (3.47)$$

The exact value of $\varepsilon$ does not affect the demonstration of the projectivity of the fractionalized operators.

We can now show that the edge modes have projective representations for $(\alpha,\beta)\neq(0,0)$:

$$K_{d,l}^{(k_1,k_2)}\tilde{X}_{l-2,l} = \omega^{-\beta}\tilde{X}_{l-2,l}K_{d,l}^{(k_1,k_2)}$$
$$K_{d,l}^{(k_1,k_2)}\check{X}_l = \omega^{\alpha+\beta}\check{X}_l K_{d,l}^{(k_1,k_2)}$$
$$K_{d,L}^{(k_1,k_2)}\tilde{X}_{L-2,L} = \omega^{-\alpha}\tilde{X}_{L-2,L}K_{d,L}^{(k_1,k_2)}$$
$$K_{d,L}^{(k_1,k_2)}\check{X}_L = \check{X}_L K_{d,L}^{(k_1,k_2)}. \qquad (3.48)$$

Therefore, $H_{c'}$ for $\alpha\neq 0$ or $\beta\neq 0$ is a new dSPT protected by the same symmetries $\{C^o,C^e,D^o,D^e,K_d\}$ as $H_d$ but distinct from it. Each integer pair $(\alpha,\beta)$ defines an SPT distinct from the dipolar cluster model as well as from each other.

## F. dSPT from holography

The Ising model and the cluster model are characterized by different symmetries, namely $\mathbb{Z}_N$ and $\mathbb{Z}_N\times\mathbb{Z}_N$. Not surprisingly, they arise as boundary theories of the single and two coupled layers of the toric codes, respectively. The dipolar Ising model and the dSPT model given in (3.5), on the other hand, share the *same* charge and dipole symmetries given in (3.2), suggesting that both models may emerge from the same bulk theory in the holographic framework.

We begin by identifying the proper bulk theory to be the anisotropic, dipolar toric code on a 2D square lattice proposed in [35]. It is a stabilizer code with two kinds of stabilizers:

$$H_{adTC} = -\sum_i V_i - \sum_i P_i + \text{h.c.}$$
$$V_i = X_{i+\hat{x}}X_i^{-2}X_{i-\hat{x}}X_{i+\frac{\hat{y}}{2}}X_{i-\frac{\hat{y}}{2}}^{-1}$$
$$P_i = Z_{i-\frac{\hat{y}}{2}+\hat{x}}Z_{i-\frac{\hat{y}}{2}}^{-2}Z_{i-\frac{\hat{y}}{2}-\hat{x}}Z_i Z_{i-\hat{y}}^{-1}. \qquad (3.49)$$

The vertices of the square lattice are labeled by $i$ (to be distinguished from $j$ labeling the sites of 1D lattice), while the link variables are defined at $i+\hat{y}/2$ - see Fig. 1(a).

We refer to the anyons associated with $V_i$ and $P_i$ as $e$ and $m$ anyons, respectively. Conservation laws associated with the anyons are

$$\prod_i V_i = 1 = \prod_i P_i, \qquad \prod_i V_i^{i_x} = 1 = \prod_i P_i^{i_x}, \qquad (3.50)$$

referring to the charge and the $x$-dipole moment conservations of the $e$ and $m$ anyons. A fully isotropic 2D model with both charge and dipole conservations, known as the rank-2 toric code (R2TC) has been proposed as well [63], and one can also think of the model (3.49) as its anisotropic version.

Consider placing the theory on a cylinder with periodic boundary conditions along the $x$-direction and open boundaries at $y = 1/2$ and $y = L_y - 1/2$ ($L_y \in \mathbb{N}$), with a rough boundary terminating on the $y$-links on both edges as shown in Fig. 1(b). We fix the boundary conditions at the bottom edge by introducing the truncated stabilizer and the bottom boundary Hamiltonian as [refer to Fig. 1(b)]

$$P_{i_x}^b = Z_{i_x\hat{x}+\hat{y}/2+\hat{x}}Z_{i_x\hat{x}+\hat{y}/2}^{-2}Z_{i_x\hat{x}+\hat{y}/2-\hat{x}}Z_{i_x\hat{x}+\hat{y}}$$
$$H_{bot}^{(e)} = -\sum_{i_x}P_{i_x}^b + \text{h.c.} \qquad (3.51)$$

The superscript $b$ refers to the stabilizers being defined at the bottom of the cylinder. Together, ground states of the bulk and bottom boundary Hamiltonian define the low-energy Hilbert space of the effective (1+1)D system. This choice of boundary conditions corresponds to the condensation of $e$ anyons, hence the superscript in $H_{bot}^{(e)}$.

We then identify the Wilson loop operator extended along the $y$-axis that commutes with $H_{bot}^{(e)}$:

$$W_{i_x}^{(e)} = \prod_{i_y}Z_{i-\hat{y}/2}. \qquad (3.52)$$

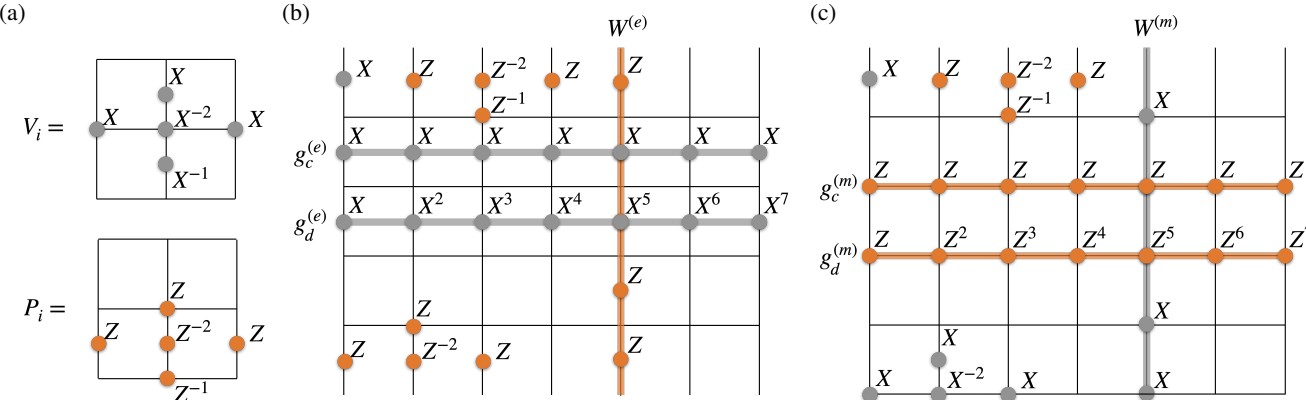

FIG. 1. (a) Bulk stabilizers of the anistropic dipolar toric code given in (3.49). (b) Boundary operators (3.51) at the bottom rough boundary fixing the *e*-condensing boundary conditions, active boundary operators at the top boundary, vertical Wilson loop operator playing the role of charge operator, and two horizontal Wilson loop operators playing the role of charge and dipole symmetry operators $g_c^{(e)}, g_d^{(e)}$ are depicted. (c) Same as (b), for the smooth bottom boundary realizing the *m*-condensing boundary condition. The top boundary is rough in both (b) and (c).

This operator already commutes with the bulk stabilizers and plays the role of a charge operator at each site $i_x$ in the effective (1+1)D theory. The Wilson loop operators extended along the *x*-axis that commute non-trivially with $W_{i_x}^{(e)}$ are

$$g_c^{(e)} = \prod_{i_x} X_{i-\hat{y}/2}, \qquad g_d^{(e)} = \prod_{i_x} X_{i-\hat{y}/2}^{i_x}. \qquad (3.53)$$

They serve as the charge and dipole symmetry operators of the effective (1+1)D system, respectively [64]. The significance of translational symmetry in the holography picture is evident from the structure of $g_d^{(e)}$ given in (3.53) which undergoes a nontrivial transformation under the action of the translation operation.

All three Wilson loop operators are depicted in Fig. 1(b). The commutation relations among $W_{i_x}^{(e)}$, $g_c^{(e)}$ and $g_d^{(e)}$ are recovered by identifying them with effective Pauli operators $(\bar{X}_j, \bar{Z}_j)$

$$W_{i_x}^{(e)} \to \bar{Z}_j \qquad g_c^{(e)} \to \prod_j \bar{X}_j \qquad g_d^{(e)} \to \prod_j \bar{X}_j^j, \qquad (3.54)$$

where we have identified $i_x$ of the 2D site $i$ with the site $j$ in the 1D chain.

Now we turn to the top boundary and write down all symmetry-allowed terms:

$$V_{i_x}^t = X_{i_x\hat{x}+(L_y-1/2)\hat{y}},$$
$$P_{i_x}^t = Z_{i_x\hat{x}+(L_y-1/2)\hat{y}+\hat{x}}Z_{i_x\hat{x}+(L_y-1/2)\hat{y}}^{-2}Z_{i_x\hat{x}+(L_y-1/2)\hat{y}-\hat{x}}Z_{i_x\hat{x}+(L_y-1)\hat{y}}^{-1},$$
$$(3.55)$$

with the superscript *t* representing their localization near the top boundary. These operators commute with the bulk stabilizers and thus act within the low-energy Hilbert space in which the bulk and the bottom-boundary stabilizers are all equal to one. In terms of the effective spins $(\bar{X}, \bar{Z})$ introduced in (3.54), we can map these top-localized operators to effective 1D spin operators:

$$V_{i_x}^t \to \bar{X}_j, \qquad P_{i_x}^t \to \bar{Z}_{j-1}\bar{Z}_j^{-2}\bar{Z}_{j+1}. \qquad (3.56)$$

There are two kinds of boundary Hamiltonians one can construct in terms of the effective spin operators that have been identified in (3.56). One of them is the dipolar Ising model

$$H = -\sum_{i_x}(V_{i_x}^t + \lambda P_{i_x}^t) + \text{h.c.}$$
$$\xrightarrow{\text{Eq. (3.56)}} -\sum_j \bar{X}_j - \lambda \sum_j \bar{Z}_{j+1}\bar{Z}_j^{-2}\bar{Z}_{j-1} + \text{h.c.}, \qquad (3.57)$$

with the transition between the paramagnetic and the symmetry-breaking phase controlled by $\lambda$. The other is the dipolar SPT model

$$H = -\sum_{i_x}\sum_{m=1}^N (\omega^{-k}V_{i_x}^t(P_{i_x}^t)^k)^m$$
$$\xrightarrow{\text{Eq. (3.56)}} -\sum_j \sum_{m=1}^N \left((\bar{Z}_{j-1}\bar{Z}_j^\dagger)^k \bar{X}_j(\bar{Z}_j^\dagger \bar{Z}_{j+1})^k\right)^m. \qquad (3.58)$$

As anticipated, both the dipolar Ising model and the dipolar SPT model share the same symmetry and can arise as boundary theories of the same 2D bulk model.

Instead of rough boundary, consider the case of smooth bottom boundary terminating on the vertices as shown in Fig. 1(c). Physically, this corresponds to having an *m*-anyon condensing Hamiltonian at the bottom:

$$H_{\text{bot}}^{(m)} = -\sum_{i_x}V_{i_x}^b + \text{h.c.},$$
$$V_{i_x}^b = X_{(i_x+1)\hat{x}}X_{i_x\hat{x}}^{-2}X_{(i_x-1)\hat{x}}X_{i_x\hat{x}+\hat{y}/2}. \qquad (3.59)$$

The effective charge and symmetry operators in the case of the smooth bottom boundary are [Fig. 1(c)]

$$W_{i_x}^{(m)} = \prod_{i_y}X_i \qquad g_c^{(m)} = \prod_{i_x}Z_i \qquad g_d^{(m)} = \prod_{i_x}Z_i^{i_x}, \qquad (3.60)$$

which can be identified with effective spins as

$$W_{i_x}^{(m)} \to \bar{Z}_j^{-1} \qquad g_c^{(m)} \to \prod_j \bar{X}_j \qquad g_d^{(m)} \to \prod_j \bar{X}_j^j. \qquad (3.61)$$

The symmetry-allowed operators at the (rough) top boundary, introduced in (3.55), now map to

$$V_{i_x}^t \to \bar{Z}_{j-1}\bar{Z}_j^{-2}\bar{Z}_{j+1}, \quad P_{i_x}^t \to \bar{X}_j^{-1}. \quad (3.62)$$

Comparing this with the earlier identification (3.56) in the case of rough bottom boundary, we conclude that changing the bottom boundary condition from rough to smooth, or from $e$-condensing to $m$-condensing, effectively corresponds to performing the dipolar Kramers-Wannier duality $K_{\text{dKW}}$ introduced in (3.3):

$$\bar{X}_j \to \bar{Z}_{j-1}\bar{Z}_j^{-2}\bar{Z}_{j+1}$$
$$\bar{Z}_{j-1}\bar{Z}_j^{-2}\bar{Z}_{j+1} \to \bar{X}_j^{-1}. \quad (3.63)$$

The scheme discussed so far can be generalized to produce boundary theory corresponding to the dipolar SPT model (3.34) protected by two charge and two dipole symmetries. The appropriate 2D bulk theory is that of two coupled layers of anisotropic dipolar toric code model we have already discussed. The Hamiltonian for each layer is given by

$$H_{1(2)} = -\sum_i V_{1(2),i} - \sum_i P_{1(2),i} + \text{h.c.}, \quad (3.64)$$

where the subscript 1(2) labels the layer. We introduce the $e$-condensing boundary condition for both layers at the bottom

$$H_{\text{bot}}^{(e)} = -\sum_{i_x}(P_{1,i_x}^b + P_{2,i_x}^b) + \text{h.c.}, \quad (3.65)$$

with $P_{1(2),i_x}^b$ defined in the same manner as in (3.51).

Accordingly, the symmetry-allowed Hamiltonian at the top boundary becomes

$$H = -\sum_{i_x}(V_{1,i_x}^t P_{2,i_x}^t + V_{2,i_x}^t P_{1,i_x}^t) + \text{h.c.}, \quad (3.66)$$

with $V_{1(2),i_x}^t$ and $P_{1(2),i_x}^t$ defined in the same manner as in (3.55). Via the identification:

$$V_{1,i_x}^t \to \bar{X}_{2j}, \qquad P_{1,i_x}^t \to \bar{Z}_{2j+2}\bar{Z}_{2j}^{-2}\bar{Z}_{2j-2}$$
$$V_{2,i_x}^t \to \bar{X}_{2j+1}, \qquad P_{2,i_x}^t \to \bar{Z}_{2j+3}\bar{Z}_{2j+1}^{-2}\bar{Z}_{2j-1}, \quad (3.67)$$

the model in (3.66) maps exactly to the dSPT model in (3.7) with two charge and two dipole symmetries. The layer index turns into the even/odd sublattice sites of the effective 1D model.

It was shown that the dSPT phase possesses a noninvertible symmetry $K_d = \left(\prod_j \text{SWAP}_{2j,2j+1}\right)K_{\text{dKW}}^o(K_{\text{dKW}}^e)^\dagger$. This symmetry operation is performed by first converting the rough bottom boundaries of both layers into smooth boundaries, followed by applying SWAP gates between sites with identical coordinates across the two layers, which effectively exchanges even and odd sites.

## IV. EXPONENTIAL SYMMETRY

Having considered the gauging of charge and dipole symmetries and the NIMSPT phases associated with them, we move to consider the exponentially modulated symmetry and its associated NIMSPT called eSPT. A holographic interpretation of the eSPT in terms of the bulk topological model is given.

### A. Gauging and noninvertible exponential symmetry

The exponential symmetry operator is defined as

$$E = \prod_j X_j^{a^j}, \quad (4.1)$$

When $a$ and $N$ are coprime and $a \neq 0 \pmod{\text{rad}(N)}$ [2, 36, 37], the exponential symmetry operator $E$ defined in (4.1) is a $\mathbb{Z}_N$ symmetry. Conversely, if $a = 0 \pmod{\text{rad}(N)}$, there exists an integer $m \in \mathbb{Z}$ with $a^m = 0 \pmod{N}$. In this case, the exponential symmetry acts only on the sites with $|r| < m$, and the generator $G$ becomes local in the thermodynamic limit. Likewise, when $a$ and $N$ are not coprime, the exponential symmetry reduces to a $\mathbb{Z}_q$ symmetry, where $q = N/\gcd(a,N)$.

The exponential gauge symmetry operator is defined

$$g_j = X_j\bar{Z}_{j-1}\bar{Z}_j^{-a}, \quad (4.2)$$

which generalizes the charge-gauging operator in (2.4). The global symmetry can be expressed $E = \prod_j(g_j)^{a^j}$.

The matter-field operators that commute with $E$ are $X_j$ and $Z_j^{-a}Z_{j+1}$, which transform under the exponential gauging as

$$X_j \to \bar{Z}_{j-1}^\dagger\bar{Z}_j^a, \quad Z_j^{-a}Z_{j+1} \to \bar{X}_j. \quad (4.3)$$

The exponential Kramers-Wannier (eKW) operator $K_{\text{eKW}}$ that performs this transformation is given by

$$K_{\text{eKW}} = \sum_{\mathbf{g},\mathbf{g}'}\omega^{\Sigma_j(ag_{j-1}-g_j)g'_{j-1}}|\mathbf{g}'\rangle\langle\mathbf{g}|. \quad (4.4)$$

Another exponential symmetry and the associated gauging operator can be constructed as

$$E' = \prod_j X_j^{a^{-j}} = \prod_j(g'_j)^{a^{-j}}$$
$$g'_j = X_j\bar{Z}_{j-1}^a\bar{Z}_j^{-1}. \quad (4.5)$$

Under the second eKW operator

$$K'_{\text{eKW}} = \sum_{\{g_j\},\{g'_j\}}\omega^{\Sigma_j(g_{j-1}-ag_j)g'_{j-1}}|\mathbf{g}'\rangle\langle\mathbf{g}|, \quad (4.6)$$

one obtains the transformation

$$X_j \xrightarrow{K'_{\text{eKW}}} \bar{Z}_{j-1}^{-a}\bar{Z}_j, \quad Z_j^\dagger Z_{j+1}^a \xrightarrow{K'_{\text{eKW}}} \bar{X}_j. \quad (4.7)$$

### B. Exponential SPT

The exponential cluster state representing the exponential SPT (eSPT) order is given by [2]

$$H_e = -\sum_j(Z_{2j-1}^a X_{2j} Z_{2j+1}^\dagger + Z_{2j-2}^\dagger X_{2j-1} Z_{2j}^a) + \text{h.c.} \quad (4.8)$$

with positive integer $a > 1$. The $a = 1$ case corresponds to the cSPT phase discussed in Section II. The Hamiltonian commutes with two exponential symmetry operators:

$$E^o = \prod_{j=1}(X_{2j-1})^{a^j}, \ \ E^e = \prod_{j=1}(X_{2j})^{a^{-j}}. \tag{4.9}$$

Here, exponential symmetries are modulated in the sense that translation results in non-trivial modification of the symmetry operator: $T^2 E^{o(e)} T^{-2} \neq E^{o(e)}$.

Some comments on the boundary conditions are in order. We assume that $a$ and $N$ are coprime, so that $a^{-1}$ is well-defined as an integer that, when multiplied by $a$, equal 1 mod $N$: $a \cdot a^{-1} \bmod N = 1$. For instance, when $N = 5$ and $a = 2$, $a^{-1} = 3$, since $2 \times 3 = 1 \pmod 5$. On an infinite lattice, when $a$ and $N$ are coprime, the exponential symmetry defined in (4.1) corresponds to a $\mathbb{Z}_N$ symmetry. Otherwise, it should be reduced to a $\mathbb{Z}_q$ symmetry, where $q = N/\gcd(a,N)$, modifying the symmetry-breaking conditions and GSD accordingly. If we instead impose PBC with the system size (even) $L$, the symmetry breaking GSD situation depends on whether $a^{L/2} = 1 \pmod N$. When $a^{L/2} \equiv 1 \bmod N$, the two exponential symmetries are well-defined on the closed manifold, and the resulting theory resembles a $\mathbb{Z}_N \times \mathbb{Z}_N$ symmetry-breaking phase, exhibiting a ground state degeneracy of $N^2$. However, if $a^{L/2} \not\equiv 1 \bmod N$, the exponential $\mathbb{Z}_N$ symmetry must be reduced to a $\mathbb{Z}_k$ symmetry, where $k = \gcd(a^{L/2} - 1, N)$, in order to be consistent with the PBC.

The ground state of the Hamiltonian (4.8) is

$$|\psi_e\rangle \propto \sum_{\mathbf{g}} \omega^{\sum_j g_{2j}(ag_{2j-1} - g_{2j+1})}|\mathbf{g}\rangle$$
$$= \sum_{\mathbf{g}} \omega^{\sum_j g_{2j-1}(ag_{2j} - g_{2j-2})}|\mathbf{g}\rangle, \tag{4.10}$$

which follows from applying the unitary operator

$$U_e = \prod_j CZ^a_{2j,2j-1} CZ^\dagger_{2j,2j+1}$$
$$= \prod_j CZ^\dagger_{2j-1,2j-2} CZ^a_{2j-1,2j} \tag{4.11}$$

on the product state $|+\rangle = \prod_j |+\rangle_j$. It can be shown that $U_e$ implements

$$X_{2j} \xrightarrow{U_e} Z^a_{2j-1} X_{2j} Z^\dagger_{2j+1},$$
$$X_{2j-1} \xrightarrow{U_e} Z^\dagger_{2j-2} X_{2j-1} Z^a_{2j}, \tag{4.12}$$

which results in the mapping $-\sum_j (X_j + X_j^\dagger) \xrightarrow{U_e} H_e$.

The eSPT model exhibits, in addition to the two modulated charge symmetries in (4.9), a noninvertible symmetry

$$K_e = T(K_{eKW})^o (K'_{eKW})^e, \tag{4.13}$$

where $K^o_{eKW}$ and $(K'_{eKW})^e$ are the eKW operators in (4.4) and (4.6) acting on the odd and even sublattices, respectively. It

performs the transformation

$$X_{2j} \xrightarrow{K_e} Z^{-a}_{2j-1} Z_{2j+1}$$
$$X_{2j+1} \xrightarrow{K_e} Z^\dagger_{2j} Z^a_{2j+2}$$
$$Z^\dagger_{2j} Z^a_{2j+2} \xrightarrow{K_e} X_{2j+1}$$
$$Z^{-a}_{2j-1} Z_{2j+1} \xrightarrow{K_e} X_{2j}, \tag{4.14}$$

and preserves $H_e$ [65]. The symmetry operators of the eSPT model span the fusion algebra:

$$E^e K_e = K_e E^e = K_e = E^o K_e = K_e E^o$$
$$K_e^\dagger K_e = \left(\sum_k (E^e)^k\right)\left(\sum_k (E^o)^k\right). \tag{4.15}$$

### C. Kennedy-Tasaki transformation

The KT transformation for the eSPT model is implemented by $KT_e = U_e K_e U_e^\dagger$. One can show

$$X_{2j} \xrightarrow{KT_e} X_{2j},$$
$$X_{2j+1} \xrightarrow{KT_e} X^\dagger_{2j+1},$$
$$Z^\dagger_{2j} Z^a_{2j+2} \xrightarrow{KT_e} Z^\dagger_{2j} X_{2j+1} Z^a_{2j+2},$$
$$Z^{-a}_{2j-1} Z_{2j+1} \xrightarrow{KT_e} Z^a_{2j-1} X_{2j} Z^\dagger_{2j+1}. \tag{4.16}$$

The symmetry operators of the exponential cluster model transform as

$$E^o \xrightarrow{KT_e} (E^o)^\dagger, \qquad E^e \xrightarrow{KT_e} E^e. \tag{4.17}$$

On the other hand, the noninvertible symmetry $K_e$ under the KT-conjugation becomes

$$KT_e K_e = (U_e P^o) KT_e \equiv V_e \cdot KT_e. \tag{4.18}$$

In summary, the symmetries of the exponential cluster model becomes

$$\{E^o, E^e, K_e\} \xrightarrow{KT_e} \{(E^o)^\dagger, E^e, V_e\}, \tag{4.19}$$

all of which are invertible, unitary symmetries. Since $V_e^2 = 1$, the symmetry group generated by $V_e$ is $\mathbb{Z}_2$. The overall symmetry group is described as

$$\mathbb{Z}_N^{e,e} \times (\mathbb{Z}_N^{e,o} \rtimes \mathbb{Z}_2^{V_e}). \tag{4.20}$$

Conjugating the exponential cluster model $H_e$ by $KT_e$ results in two copies of $\mathbb{Z}_N$ exponential Ising model, or double exponential Ising model (eIM2):

$$\hat{H}_e = -\sum_j \left(Z^{-a}_{2j-1} Z_{2j+1} + Z^\dagger_{2j} Z^a_{2j+2}\right) + h.c.. \tag{4.21}$$

One can explicitly check that eIM2 possesses all the symmetries shown on the right-hand side of (4.19). The ground state is characterized by

$$ag_{2j-1} = g_{2j+1}, \qquad g_{2j} = ag_{2j+2}. \tag{4.22}$$

We have altogether two quantum numbers $g_e^o, g_e^e \in \mathbb{Z}_N$ to characterize the ground state $|g_e^o, g_e^e\rangle$, where $g_{2j-1} = g_e^o \cdot a^j$ and $g_{2j} = g_e^e \cdot a^{-j}$.

It can be shown that

$$(E^o)^{\eta_1}(E^e)^{\eta_2}V_e^{\eta_3}|g_e^o, g_e^e\rangle = |\eta_1 + (-1)^{\eta_3}g_e^o, \eta_2 + g_e^e\rangle. \quad (4.23)$$

When

$$\eta_1 = 2g_e^o \ (\text{mod } N), \quad \eta_2 = 0, \quad \eta_3 = 1, \quad (4.24)$$

the ground state $|g_e^o, g_e^e\rangle$ is invariant under the operation by $(E^o)^{\eta_1}(E^e)^{\eta_2}V_e$. The order parameters are $\sum_j (Z_{2j})^{a^j}$ and $\sum_j (Z_{2j-1})^{a^{-j}}$.

### D. Other eSPT states

The SPT classification for two exponential symmetries $(E^o, E^e)$ goes as $H^2(\mathbb{Z}_N^2, U(1)) = \mathbb{Z}_N$, and represented by models

$$H_e^{(k)} = -\sum_j [X_{2j}(Z_{2j-1}^a Z_{2j+1}^\dagger)^k + X_{2j-1}(Z_{2j-2}^\dagger Z_{2j}^a)^k] + \text{h.c.}, \quad (4.25)$$

where $k \in \mathbb{Z}_N$. Only the $k = 1$ model maintains the NIS under $K_e$.

A Hamiltonian sharing the same $\{E^o, E^e, V_e\}$ symmetries as eIM2 can be constructed. Using the transformation:

$$Y_{2j} \xrightarrow{V_e} Z_{2j-1}^a Y_{2j} Z_{2j+1}^\dagger,$$

$$Y_{2j+1} \xrightarrow{V_e} Z_{2j} Y_{2j+1}^\dagger Z_{2j+2}^{-a},$$

$$Z_{2j} \xrightarrow{V_e} Z_{2j},$$

$$Z_{2j+1} \xrightarrow{V_e} Z_{2j+1}^\dagger, \quad (4.26)$$

we get

$$Y_{2j-1}^{-a} Y_{2j+1} \xrightarrow{V_e} Z_{2j-2}^{-a} Y_{2j-1}^a Z_{2j}^{a^2+1} Y_{2j+1}^\dagger Z_{2j+2}^{-a}. \quad (4.27)$$

The following model is invariant under the same set of symmetries $\{E^o, E^e, V_e\}$ as the eIM2:

$$\hat{H}_{e'} = -\sum_j \omega^{a^j\alpha} Z_{2j-2}^\dagger Z_{2j}^{-a}$$
$$- \sum_j Y_{2j-1}^{-a} Y_{2j+1} \left(1 + Z_{2j-2}^\dagger Z_{2j}^{-a^2-1} Z_{2j+2}^a\right) + \text{h.c.}, \quad (4.28)$$

where the parameter $\alpha$ satisfies $\alpha \in \mathbb{Z}_N$. All the terms in the Hamiltonian $H$ mutually commute. The first term is minimized by $Z_{2j-2}^\dagger Z_{2j}^a = \omega^{a^j\alpha}$. The product of $Z$'s in the second term becomes

$$Z_{2j-2}^a Z_{2j}^{-a^2-1} Z_{2j+2}^a = (Z_{2j-2}^\dagger Z_{2j}^a)^{-a} \cdot (Z_{2j}^\dagger Z_{2j+2}^a) \quad (4.29)$$

and equals one in the ground state. The overall ground state conditions

$$Z_{2j-2}^\dagger Z_{2j}^a = \omega^{a^j\alpha}, \quad Y_{2j-1}^{-a} Y_{2j+1} = 1 \quad (4.30)$$

results in the ground state configuration

$$g_{2j+1}^Y = ag_{2j-1}^Y, \quad g_{2j-2} = ag_{2j} - a^j\alpha, \quad (4.31)$$

and the ground states can be expressed as $|(g_e^Y)^o, (g_e)^e\rangle$, where

$$g_{2j-1}^Y = a^j(g_e^Y)^o,$$
$$g_{2j} = a^{-j}g_e^e + \alpha(a^{j-2} - a^{-j-2})(1 - a^{-2})^{-1}. \quad (4.32)$$

The inverse $(1 - a^{-2})^{-1}$ is understood as an integer that, when multiplied with $1 - a^{-2}$, equals 1 mod $N$. Since

$$\omega^{\eta_1 a^j} Z_{2j-2}^\dagger Y_{2j-1}^\dagger Z_{2j}^a \xrightarrow{(E^o)^{\eta_1}(E^e)^{\eta_2}V_e} Y_{2j-1}$$
$$\omega^{\eta_2 a^{-j}} Z_{2j} \xrightarrow{(E^o)^{\eta_1}(E^e)^{\eta_2}V_e} Z_{2j}, \quad (4.33)$$

we conclude

$$(E^o)^{\eta_1}(E^e)^{\eta_2}V_e^{\eta_3}|(g_e^Y)^o, g_e^e\rangle$$
$$= |\eta_1 + \alpha + (-1)^{\eta_3}(g_e^Y)^o, \eta_2 + g_e^e\rangle. \quad (4.34)$$

When

$$\eta_1 = 2(g_e^Y)^o - \alpha \ (\text{mod } N), \quad \eta_2 = 0, \quad \eta_3 = 1, \quad (4.35)$$

the ground state $|(g_e^Y)^o, g_e^e\rangle$ remains invariant under the action by $(E^o)^{\eta_1}(E^e)^{\eta_2}V_e$. The order parameters for this state are

$$\sum_j Z_{2j}^{a^j} \omega^{-\alpha(a^{2j-2} - a^{-2})(1-a^{-2})^{-1}}, \quad \sum_j Y_{2j-1}^{a^{-j}}. \quad (4.36)$$

Applying $\text{KT}_e^\dagger$ on the Hamiltonian (4.28) gives $\hat{H}_{e'} \xrightarrow{\text{KT}_e^\dagger} H_{e'}$ where

$$H_{e'} = -\sum_j \omega^{-a^j\alpha} Z_{2j-2}^\dagger X_{2j-1} Z_{2j}^a$$
$$- \sum_j Y_{2j-1}^a X_{2j} Y_{2j+1}^\dagger \left(1 + Z_{2j-2}^a X_{2j-1}^{-a} Z_{2j}^{-a^2-1} X_{2j+1} Z_{2j+2}^a\right)$$
$$+ \text{h.c.} \quad (4.37)$$

which preserves the same $\{E^o, E^e, K_e\}$ symmetries as the original eSPT model $H_e$. The ground state of the new eSPT model $H_{e'}$ is fixed by

$$Z_{2j-2}^\dagger X_{2j-1} Z_{2j}^a = \omega^{a^j\alpha}, \quad Y_{2j-1}^a X_{2j} Y_{2j+1}^\dagger = 1. \quad (4.38)$$

The Hamiltonian (4.37) can be simplified through two unitary rotations: $U_e$ in (4.11) and

$$W_e = \prod_j CZ_{2j-2,2j}^a CZ_{2j,2j}^{(-a^2-2+a_2)/2}, \quad (4.39)$$

where $a_2 \equiv a \mod 2$ is introduced to ensure that the exponent of $CZ_{2j,2j}$ remains an integer for arbitrary $a$. The $W_e$ operation transforms the trivial Hamiltonian $-\sum_j X_{2j}$ to a cluster model defined on the even sublattice:

$$-\sum_j X_{2j} + \text{h.c.} \xrightarrow{W_e}$$
$$-\sum_j Z_{2j-2}^a Z_{2j}^{(-a^2-2+a_2)/2} X_{2j} Z_{2j}^{(-a^2-2+a_2)/2} Z_{2j+2}^a + \text{h.c.} \quad (4.40)$$

Under the combined operation $U_{ee}^\dagger = W_e^\dagger U_e^\dagger$,

$$Z_{2j-2}^\dagger X_{2j-1} Z_{2j}^a \xrightarrow{U_{ee}^\dagger} X_{2j-1},$$

$$Y_{2j-1}^a X_{2j} Y_{2j+1}^\dagger \xrightarrow{U_{ee}^\dagger} \omega^{(1-a_2)/2} X_{2j-1}^a X_{2j} Z_{2j}^{1-a_2} X_{2j+1}^\dagger, \quad (4.41)$$

and the Hamiltonian $H_{e'}$ transforms to

$$H_{e'} \xrightarrow{U_{ee}^\dagger} -\sum_j \omega^{-a^j\alpha} X_{2j-1} - \sum_j \omega^{(1-a_2)/2} X_{2j-1}^a X_{2j} Z_{2j}^{1-a_2} X_{2j+1}^\dagger$$
$$- \sum_j \omega^{(1-a_2)/2} X_{2j} Z_{2j}^{1-a_2} + \text{h.c.} \quad (4.42)$$

This Hamiltonian is not written entirely in terms of $X$ operators, but one can still easily identify its ground state. Depending on $a$ being odd or even ($a_2 = 1$ or $0$), the ground state is characterized by

$$g_{2j-1}^X = a^j\alpha, \qquad g_{2j}^X = 1 \qquad (a_2 = 1)$$
$$g_{2j-1}^X = a^j\alpha, \qquad g_{2j}^Y = 1 \qquad (a_2 = 0). \quad (4.43)$$

---

Explicitly, the ground state of the new eSPT Hamiltonian $H_{e'}$ in (4.37) is written

$$W_e U_e |(g^X)^o = \alpha, (g^X)^e = 1\rangle \qquad (a_2 = 1)$$
$$W_e U_e |(g^X)^o = \alpha, (g^Y)^e = 1\rangle \qquad (a_2 = 0), \quad (4.44)$$

where $g_{2j-1}^X = a^j(g^X)^o$, $g_{2j}^X = (g^X)^e$, and $g_{2j}^Y = (g^Y)^e$. Again the new eSPT allows a two-layer DDW interpretation.

## E. Edge modes

We now consider the Hamiltonian $H_{e|e'}$:

$$H_{e|e'} = -\sum_{j=1}^{l/2} Z_{2j-2}^\dagger X_{2j-1} Z_{2j}^a - \sum_{j=1}^{l/2-1} Z_{2j-1}^a X_{2j} Z_{2j+1}^\dagger - \sum_{j=l/2+1}^{L/2} \omega^{-a^j\alpha} Z_{2j-2}^\dagger X_{2j-1} Z_{2j}^a$$
$$- \sum_{j=l/2+1}^{L/2-1} Y_{2j-1}^a X_{2j} Y_{2j+1}^\dagger \left(1 + Z_{2j-2}^a X_{2j-1}^{-a} Z_{2j}^{-a^2-1} X_{2j+1} Z_{2j+2}^a\right) + \text{h.c..} \quad (4.45)$$

This is a model in which $H_e$ acts on the sites $1 \le j \le l$ (with even $l$), and $H_{e'}$ acts on the sites $l+1 \le j \le L$ (with even $L$), where $L$ and $l$ are chosen to satisfy

$$a^l = 1 \qquad (\text{mod } N) \quad (4.46)$$

A periodic boundary condition is imposed. The ground states are required to satisfy

$$Z_{2j-2}^\dagger X_{2j-1} Z_{2j}^a = 1 \qquad (1 \le j \le l/2)$$
$$Z_{2j-1}^a X_{2j} Z_{2j+1}^\dagger = 1 \qquad (1 \le j \le l/2-1)$$
$$Z_{2j-2}^\dagger X_{2j-1} Z_{2j}^a = \omega^{a^j\alpha} \quad (l/2+1 \le j \le L/2)$$
$$Y_{2j-1}^a X_{2j} Y_{2j+1}^\dagger = 1 \qquad (l/2+1 \le j \le L/2-1). \quad (4.47)$$

From these conditions, one can derive

$$E^o|\psi\rangle = |\psi\rangle, \qquad E^e|\psi\rangle = \tilde{X}_l \tilde{X}_L |\psi\rangle,$$
$$Z_l \tilde{X}_l = \omega^{a^{-l/2}} \tilde{X}_l \tilde{Z}_l, \qquad Z_L \tilde{X}_L = \omega \tilde{X}_L \tilde{Z}_L, \quad (4.48)$$

where

$$\tilde{X}_l = Z_{l-1}^{a^{-l/2+1}} X_l^{a^{-l/2}} Y_{l+1}^{-a^{-l/2}} \qquad \tilde{X}_L = Y_{L-1}^a X_L Z_1^{-1}. \quad (4.49)$$

The action of $K_e$ within the ground state subspace can found by examining how $K_e$ acts on the fractionalized operators in

(4.49) within the ground state subspace [22]:

$$\tilde{X}_l K_e |\psi\rangle = \omega^{-a\alpha} K_e \tilde{X}_l^\dagger |\psi\rangle$$
$$\tilde{X}_L K_e |\psi\rangle = \omega^{a\alpha} K_e \tilde{X}_L^\dagger |\psi\rangle$$
$$Z_l^\dagger Z_L^{a^{-l/2}} K_e |\psi\rangle = K_e Z_l Z_L^{-a^{-l/2}} |\psi\rangle$$
$$K_e^2 |\psi\rangle = N \sum_{k=1}^N (\tilde{X}_l \tilde{X}_L)^k |\psi\rangle. \quad (4.50)$$

An explicit expression for $K_e$ that satisfies these relations is given by:

$$K_e |\psi\rangle = \varepsilon P Z_l^{-a^{-l/2+1}\alpha} Z_L^{a\alpha} \left(\sum_{k=1}^N (\tilde{X}_l \tilde{X}_L)^k\right) |\psi\rangle$$
$$= \varepsilon P \left(\sum_{k=1}^N K_{e,l}^{(k)} K_{e,L}^{(k)}\right) |\psi\rangle, \quad (4.51)$$

where we introduce the abbreviations

$$K_{e,l}^{(k)} = Z_l^{-a^{-l/2+1}\alpha} \tilde{X}_l^k, \quad K_{e,L}^{(k)} = Z_L^{a\alpha} \tilde{X}_L^k$$

in the second line. The prefactor $\varepsilon$ is either $+1$ or $-1$.

Finally, we demonstrate that the fractionalized KW operators satisfy the algebra

$$K_{e,l}^{(k)} \tilde{X}_l = \omega^{-a\alpha} \tilde{X}_l K_{e,l}^{(k)}, \qquad K_{e,L}^{(k)} \tilde{X}_L = \omega^{a\alpha} \tilde{X}_L K_{e,L}^{(k)} \quad (4.52)$$

thereby confirming that they furnish projective representations for all $\alpha \neq 0$. Other conclusions derived for cSPTs can be readily generalized to the case of eSPTs.

### F. eSPT from holography

In this section, we present a holographic perspective on eSPT by interpreting them as boundaries of 2D topological orders obtained from gauging the exponential symmetries introduced in [36]. Notably, the noninvertible symmetry defined in (4.4) can be understood as arising from creating a genon defect at the edge of these topological ordered states.

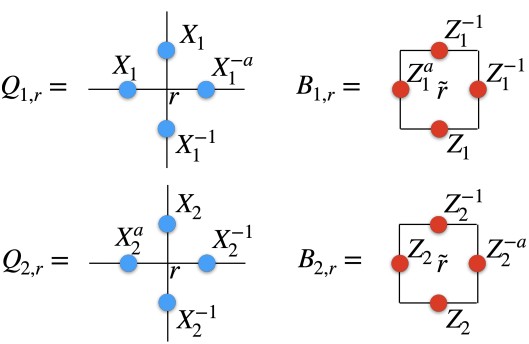

FIG. 2. Stabilizers for each layer of eTC. The layers are labeled by 1,2. $r$ and $\tilde{r}$ represent the vertex and plaquette coordinates.

The construction relies on having two layers of exponentially modulated toric codes obtained by gauging exponential symmetries. Each modulated toric code consists of a pair of stabilizers shown in Fig. 2. Such model was introduced in [36] and will be referred to as exponential toric code (eTC). In Fig. 2, the exponent $a$ is introduced to induce modulation of the Wilson loop operator along the $x$-axis, while preserving uniformity along the $y$-axis. The original model [36] introduced modulations along both axes, but for our purpose only the modulation along the $x$-direction is needed. The edge mode arising at the boundary of a single layer of eTC is the exponentially modulated Ising model [66].

The eTC Hamiltonian for each layer labeled by subscript 1,2 is

$$H_{1(2)} = -\sum_r Q_{1(2),r} - \sum_{\tilde{r}} B_{1(2),\tilde{r}} + \text{h.c.}, \tag{4.53}$$

where

$$
\begin{aligned}
Q_{1,r} &= X_{1,r+\frac{\hat{x}}{2}}^{-a} X_{1,r-\frac{\hat{x}}{2}} X_{1,r+\frac{\hat{y}}{2}} X_{1,r-\frac{\hat{y}}{2}}^{-1}, \\
Q_{2,r} &= X_{2,r+\frac{\hat{x}}{2}}^{-1} X_{2,r-\frac{\hat{x}}{2}}^{a} X_{2,r+\frac{\hat{y}}{2}} X_{2,r-\frac{\hat{y}}{2}}^{-1}, \\
B_{1,\tilde{r}} &= Z_{1,\tilde{r}+\frac{\hat{x}}{2}}^{-1} Z_{1,\tilde{r}-\frac{\hat{x}}{2}}^{a} Z_{1,\tilde{r}+\frac{\hat{y}}{2}}^{-1} Z_{1,\tilde{r}-\frac{\hat{x}}{2}}, \\
B_{2,\tilde{r}} &= Z_{2,\tilde{r}+\frac{\hat{x}}{2}}^{-a} Z_{2,\tilde{r}-\frac{\hat{x}}{2}} Z_{2,\tilde{r}+\frac{\hat{y}}{2}}^{-1} Z_{2,\tilde{r}-\frac{\hat{y}}{2}}.
\end{aligned}
\tag{4.54}
$$

Coordinates denoted by $r$ ($\tilde{r}$) refer to the vertex (plaquette center) of the square lattice, as shown in Fig. 2. These coordinates are related by $r + \frac{\hat{y}}{2} = \tilde{r} + \frac{\hat{x}}{2}$.

The ground states of the above Hamiltonian are projected onto the vanishing charge and flux sector by the $Q_r$ and $B_{\tilde{r}}$ operators. As discussed in [36], the model realizes either topologically ordered phases or trivial phases depending on the parameters $a$. The Hamiltonian embodies the following conservation laws:

$$
\begin{aligned}
\prod_r (Q_{1,r})^{a^{r_x}} &= 1, & \prod_{\tilde{r}} (B_{1,\tilde{r}})^{a^{-r_x}} &= 1, \\
\prod_r (Q_{2,r})^{a^{-r_x}} &= 1, & \prod_{\tilde{r}} (B_{2,\tilde{r}})^{a^{r_x}} &= 1.
\end{aligned}
\tag{4.55}
$$

The theory thus describes a modulated gauge theory with the exponential charge and flux being conserved.

Suppose we place such a bilayer system on a stripe that is periodic along the $x$ direction with open boundary at $y = 0$ and $y = L$, as depicted in Fig. 3. We can choose boundary stabilizers on the *rough boundary* at the bottom ($y = 0$) as

$$
\begin{aligned}
H_{\text{bot}} = &- \sum_{\tilde{r} \in \text{edge}} Z_{2,\tilde{r}+\frac{\hat{x}}{2}}^{-a} Z_{2,\tilde{r}-\frac{\hat{x}}{2}} Z_{2,\tilde{r}+\frac{\hat{y}}{2}}^{-1} \\
&- \sum_{\tilde{r} \in \text{edge}} Z_{1,\tilde{r}+\frac{\hat{x}}{2}}^{-1} Z_{1,\tilde{r}-\frac{\hat{x}}{2}}^{a} Z_{1,\tilde{r}+\frac{\hat{y}}{2}}^{-1} + \text{h.c..}
\end{aligned}
\tag{4.56}
$$

These boundary stabilizers, shown in Fig. 3(a), take on the value +1 along with the bulk stabilizers. Such boundary stabilizer allows the exponential charge to be condensed at the boundary. The bottom boundary defines the holonomies of the stripe, which label the different degenerate ground states. When the holonomy operators are pushed to the top boundary, they can also be interpreted as a global exponential symmetry acting on the 1D edge. We have two such holonomy operators in the bilayer eTC, one from each layer:

$$g_1 = \prod_{\tilde{r}_x} X_{1,\tilde{r}+\frac{\hat{x}}{2}}^{a^{\tilde{r}_x}}, \qquad g_2 = \prod_{\tilde{r}_x} X_{2,\tilde{r}+\frac{\hat{x}}{2}}^{a^{-\tilde{r}_x}}, \tag{4.57}$$

where $\tilde{r}_x$ refers to the $x$-coordinate of $\tilde{r}$. One can observe that the two Wilon operators $g_1$ and $g_2$ undergo nontrivial transformations under the action of the translation operation. Additionally, the Wilson loop operators extended along the $y-$axis that do not commute with $g_1$ or $g_2$ are

$$W_1(\tilde{r}_x) = \prod_{\tilde{r}_y} Z_{1,\tilde{r}+\frac{\hat{x}}{2}}, \qquad W_2(\tilde{r}_x) = \prod_{\tilde{r}_y} Z_{2,\tilde{r}+\frac{\hat{x}}{2}}, \tag{4.58}$$

with the product running along the $y$-coordinate of $\tilde{r}$ given by $\tilde{r}_y$. We can identify these operators with some effective spin operators:

$$
\begin{aligned}
g_1 &\to \prod_j \bar{X}_{2j+1}^{a^j}, & g_2 &\to \prod_j \bar{X}_{2j}^{a^{-j}} \\
W_1(\tilde{r}_x) &\to \bar{Z}_{2j+1}, & W_2(\tilde{r}_x) &\to \bar{Z}_{2j}.
\end{aligned}
\tag{4.59}
$$

Operators in each layer are converted to the effective 1D spin operators on the even and odd sublattice sites.

The operators at the top edge that can commute with the two symmetry operators $g_1$ and $g_2$ in (4.57) are $Z_{2,\tilde{r}+\frac{\hat{x}}{2}}^{a} Z_{2,\tilde{r}-\frac{\hat{x}}{2}}^{-1} Z_{2,\tilde{r}-\frac{\hat{y}}{2}}^{-1}$, $Z_{1,\tilde{r}+\frac{\hat{x}}{2}}^{-1} Z_{1,\tilde{r}-\frac{\hat{x}}{2}}^{a} Z_{1,\tilde{r}-\frac{\hat{y}}{2}}$, and $X_{1(2),\tilde{r}-\frac{\hat{x}}{2}}$. Rather than forming a top edge-localized Hamiltonian in terms of these operators acting

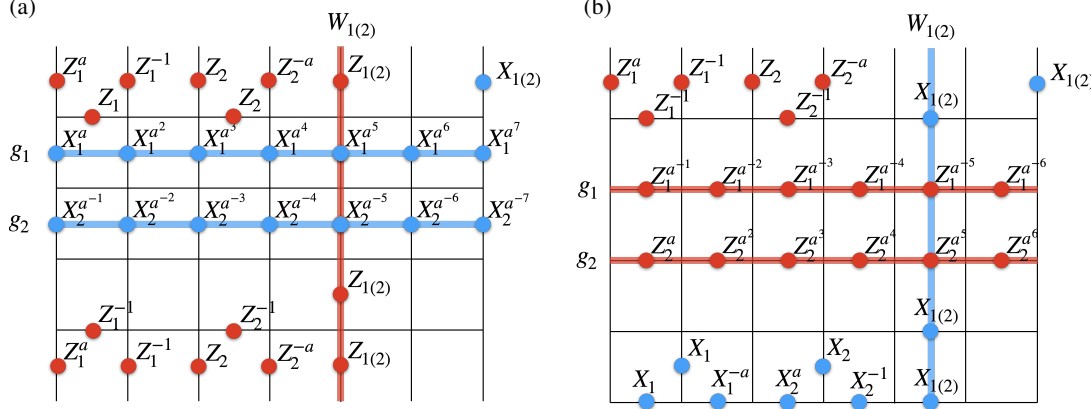

FIG. 3. (a) Boundary operators at the bottom rough boundary [Eq. (4.56)] and the top rough boundary [Eq. (4.60)], serving as the boundary condition and the active degrees of freedom, respectively, of each layer of the exponential toric codes. Two horizontal Wilson loop operators serving as two symmetry operators of the eSPT, $g_1, g_2$ for layers 1 and 2, are shown along with the vertical Wilson loop operators serving as the charge operators. (b) Similar to (a) for smooth bottom boundary and rough top boundary.

on one of the layers, one can form a composite Hamiltonian that acts on both layers at once. One such possibility is offered by the edge stabilizer model with rough boundary as shown in Fig. 3(a):

$$H_{\text{top}} = - \sum_{\tilde{r}\in\text{edge}} Z^a_{2,\tilde{r}+\frac{\hat{x}}{2}} Z^{-1}_{2,\tilde{r}-\frac{\hat{x}}{2}} Z^{-1}_{2,\tilde{r}-\frac{\hat{y}}{2}} X_{1,\tilde{r}-\frac{\hat{x}}{2}}$$
$$- \sum_{\tilde{r}\in\text{edge}} Z^{-1}_{1,\tilde{r}+\frac{\hat{x}}{2}} Z^a_{1,\tilde{r}-\frac{\hat{x}}{2}} Z_{1,\tilde{r}-\frac{\hat{y}}{2}} X_{2,\tilde{r}+\frac{\hat{x}}{2}} + \text{h.c.}. \quad (4.60)$$

Using (4.59), the terms that commute with $g_1$ and $g_2$ are identified as

$$X_{1,r+\frac{\hat{y}}{2}} \to X_{2j+1}$$
$$Z^{\dagger}_{1,\tilde{r}+\frac{\hat{x}}{2}} Z^a_{1,\tilde{r}-\frac{\hat{x}}{2}} Z^{-1}_{1,\tilde{r}-\frac{\hat{y}}{2}} \to \bar{Z}^a_{2j-1} \bar{Z}^{\dagger}_{2j+1}$$
$$X_{2,r+\frac{\hat{y}}{2}} \to X_{2j}$$
$$Z^{-a}_{2,\tilde{r}+\frac{\hat{x}}{2}} Z_{2,\tilde{r}-\frac{\hat{x}}{2}} Z_{2,\tilde{r}-\frac{\hat{y}}{2}} \to \bar{Z}_{2j-2} \bar{Z}^{-a}_{2j}, \quad (4.61)$$

and $H_{\text{top}}$ is indeed identified with the eSPT model in (4.8).

Now consider the case of a smooth bottom boundary terminating on the vertices as shown in Fig. 3(b), and the bottom boundary Hamiltonian given by

$$H_{\text{bot}} = - \sum_{r\in\text{edge}} X^{-a}_{1,r+\frac{\hat{x}}{2}} X_{1,r-\frac{\hat{x}}{2}} X_{1,r+\frac{\hat{y}}{2}}$$
$$- \sum_{r\in\text{edge}} X^{-1}_{2,r+\frac{\hat{x}}{2}} X^a_{2,r-\frac{\hat{x}}{2}} X_{2,r+\frac{\hat{y}}{2}} + \text{h.c.}. \quad (4.62)$$

The effective symmetry operators in the case of smooth bottom boundary are [Fig. 3(b)]

$$W_{1(2)}(r_x) = \prod_{r_y} X_{1(2),r+\frac{\hat{x}}{2}}, \quad g_1 = \prod_{r_x} Z^{a^{-r_x}}_{1,r+\frac{\hat{x}}{2}}, \quad g_2 = \prod_{r_x} X^{a^{r_x}}_{2,r+\frac{\hat{x}}{2}},$$

which can be identified with effective spins as

$$W_1(r_x) \to \bar{Z}^{\dagger}_{2j+1} \qquad W_2(r_x) \to \bar{Z}^{\dagger}_{2j}$$
$$g_1 \to \prod_j \bar{X}^{a^{-j}}_{2j+1} \qquad g_2 \to \prod_j \bar{X}^{a^j}_{2j}. \quad (4.63)$$

The symmetry-allowed operators at the top boundary now map to

$$X_{1,r+\frac{\hat{y}}{2}} \to Z^{\dagger}_{2j-1} Z^a_{2j+1}$$
$$Z^{\dagger}_{1,\tilde{r}+\frac{\hat{x}}{2}} Z^a_{1,\tilde{r}-\frac{\hat{x}}{2}} Z^{-1}_{1,\tilde{r}-\frac{\hat{y}}{2}} \to \bar{X}_{2j-1}$$
$$X_{2,r+\frac{\hat{y}}{2}} \to Z^{-a}_{2j-2} Z_{2j}$$
$$Z^{-a}_{2,\tilde{r}+\frac{\hat{x}}{2}} Z_{2,\tilde{r}-\frac{\hat{x}}{2}} Z_{2,\tilde{r}-\frac{\hat{y}}{2}} \to \bar{X}^{\dagger}_{2j-2}. \quad (4.64)$$

By comparing this with the earlier identification (4.61) for the case of a rough bottom boundary, we conclude that changing the boundary condition from rough to smooth at the bottom boundary effectively corresponds to applying the exponential Kramers-Wannier duality, $K_{\text{eKW}}$ at the top of the first layer and $K'_{\text{eKW}}$ on the second layer. Given that the noninvertible symmetry of the eSPT model is defined as $K_e = T(K_{\text{eKW}})^o (K'_{\text{eKW}})^e$, the process of changing the boundary condition followed by applying the sequential SWAP gates $\prod_i \text{SWAP}_{r,r+\hat{x}}$-which act on the edge and effectively implement the lattice translation $r \to r+\hat{x}$-is equivalent to performing the symmetry operation $K_e$.

## V. SUMMARY AND DISCUSSION

Given the intricate structure of our results, we give a summary of the key physical insights in the subsection below, and follow it by a detailed discussion of our findings.

### A. Summary of our findings

We identified several distinct cSPTs protected by the two charge symmetries $C^o$ and $C^e$ together with the non-invertible symmetry $K_c$. In addition to the well-known cSPT,

$$H_c = - \sum_j (Z_{2j-1} X_{2j} Z^{\dagger}_{2j+1} + Z^{\dagger}_{2j-2} X_{2j-1} Z_{2j}) + \text{h.c.}, \quad (5.1)$$

we identify two new families of cSPTs, $H_{c'}$ and $H_{c''}$, parameterized by $\alpha \in \mathbb{Z}_N$, which are given by:

$$H_{c'} = -\sum_j \omega^{-\alpha} Z_{2j}^\dagger X_{2j+1} Z_{2j+2}$$
$$-\sum_j Y_{2j-1} X_{2j} Y_{2j+1}^\dagger (1 + Z_{2j-2} X_{2j-1}^\dagger Z_{2j}^{-2} X_{2j+1} Z_{2j+2})$$
$$+ \text{h.c.}.$$
$$H_{c''} = -\sum_j \omega^{-\alpha} Z_{2j-1} X_{2j} Z_{2j+1}^\dagger$$
$$-\sum_j Y_{2j}^\dagger X_{2j+1} Y_{2j+2} (1 + Z_{2j-1}^\dagger X_{2j}^\dagger Z_{2j+1}^2 X_{2j+2} Z_{2j+3}^\dagger)$$
$$+ \text{h.c.}. \tag{5.2}$$

Similarly, we identified distinct dSPTs protected by $C^o$, $C^e$, $D^o$, $D^e$, and $K_d$. In addition to the $H_d$:

$$H_d = -\sum_j Z_{2j-1} X_{2j} Z_{2j+1}^{-2} Z_{2j+3}$$
$$-\sum_j Z_{2j-2} Z_{2j}^{-2} X_{2j+1} Z_{2j+2} + \text{h.c.}, \tag{5.3}$$

we identify one family of dSPT, $H_{d'}$, parametrized by $\alpha, \beta \in \mathbb{Z}_N$, which are given by:

$$H_{d'} = -\sum_j \omega^{\alpha+\beta j} Z_{2j-2} Z_{2j}^{-2} X_{2j+1} Z_{2j+2}$$
$$-\sum_j Y_{2j-1} X_{2j} Y_{2j+1}^{-2} Y_{2j+3}$$
$$\left(1 + Z_{2j-4}^\dagger Z_{2j-2}^4 X_{2j-1}^\dagger Z_{2j}^{-6} X_{2j+1}^2 Z_{2j+2}^4 X_{2j+3}^\dagger Z_{2j+4}^\dagger\right)$$
$$+ \text{h.c.}. \tag{5.4}$$

For the last, we identified distinct eSPTS protected by $E^o$, $E^e$, and $K_e$. In addition to the $H_e$:

$$H_e = -\sum_j (Z_{2j-1}^a X_{2j} Z_{2j+1}^\dagger + Z_{2j-2}^\dagger X_{2j-1} Z_{2j}^a) + \text{h.c.}, \tag{5.5}$$

we identify one family of eSPT, $H_{e'}$, parametrized by $\alpha \in \mathbb{Z}_N$, which are given by:

$$H_{e'} = -\sum_j \omega^{-a^j \alpha} Z_{2j-2}^\dagger X_{2j-1} Z_{2j}^a$$
$$-\sum_j Y_{2j-1}^a X_{2j} Y_{2j+1}^\dagger \left(1 + Z_{2j-2}^a X_{2j-1}^{-a} Z_{2j}^{-a^2-1} X_{2j+1} Z_{2j+2}^a\right)$$
$$+ \text{h.c.}. \tag{5.6}$$

### B. Discussion

Modulated symmetry represents an extension of the global symmetry in quantum theories, with spatially modulated symmetry charges and modified conservation laws. Recent works suggested that a noninvertible symmetry may generally coexist with one of these modulated symmetries. We have investigated this connection in several models of one-dimensional

SPTs dubbed cSPT, dSPT, and eSPT, to show how SPTs protected by modulated symmetries in general possess noninvertible symmetry as well, with interesting ramifications.

We have identified the appropriate Kramers-Wannier and Kennedy-Tasaki transformations in all three SPTs and used them to map the given SPT model and its protecting symmetries to a symmetry-breaking model and its symmetries. The enriched symmetry structure of the original SPT model due to the presence of noninvertible symmetry leads to a similarly enriched structure in the symmetry group of the dual SSB model. Taking advantage of this enlarged symmetry, we have identified some new SSB models sharing the same set of symmetries as the original SSB model. The dual of the new SSB model results in a new kind of SPT, which is distinct from the original SPT by virtue of the fractionalization of the noninvertible symmetry. This proves that modulated SPTs are indeed endowed with a richer symmetry structure than what the group-based cohomology classification would allow. While a similar investigation for the rich set of cSPTs in the presence of noninvertible symmetry has been done for $\mathbb{Z}_2$ [22] and $\mathbb{Z}_N$ [59] cluster states, our work represents the first systematic study of its kind for SPTs protected by modulated symmetries. Table I summarizes the symmetry group structures (excluding the noninvertible part) of the SPT phases and of the SSB phases related by the KT transformation.

|  | SG of SPT | SG after KT |
|---|---|---|
| cSPT | $\mathbb{Z}_N^e \times \mathbb{Z}_N^o$ | $\mathbb{Z}_N^e \times (\mathbb{Z}_N^o \rtimes \mathbb{Z}_2^{V_c})$ |
| dSPT | $\mathbb{Z}_N^{c,e} \times \mathbb{Z}_N^{d,e} \times \mathbb{Z}_N^{c,o} \times \mathbb{Z}_N^{d,o}$ | $\mathbb{Z}_N^{c,e} \times \mathbb{Z}_N^{d,e} \times [(\mathbb{Z}_N^{c,o} \times \mathbb{Z}_N^{d,o}) \rtimes \mathbb{Z}_2^{V_d}]$ |
| eSPT | $\mathbb{Z}_N^{e,e} \times \mathbb{Z}_N^{e,o}$ | $\mathbb{Z}_N^{e,e} \times (\mathbb{Z}_N^{e,o} \rtimes \mathbb{Z}_2^{V_e})$ |

TABLE I. Symmetry group (SG) of the SPT phases-excluding those associated with NIS-as well as those obtained after applying the KT transformation are summarized for the (c,d,e)SPT phases. The definition of each symmetry element can be found in the corresponding section.

In addition, we have identified the two-dimensional modulated gauge theory with topological order whose one-dimensional boundary physics precisely captures the given SPT phase. The bulk theories thus identified are two coupled layers of toric codes, of anisotropic dipolar toric codes [35], and of exponentially modulated toric codes [36] in the case of cSPT, dSPT, and eSPT, respectively. Switching the boundary from rough to smooth, or from $e$-condensing to $m$-condensing boundary condition, has the same effect as performing the Kramers-Wannier transformation in accordance with the general scheme of topological holography [21]. A holographic interpretation of 1D SPT phases protected by non-modulated symmetries was put forward earlier in [67], and those of SPT-trivial models with modulated symmetries in [68]. Holographic considerations for explicit examples of modulated SPTs are given here for the first time - see also the related upcoming article [69].

One-dimensional SPT models protected by modulated symmetries have been proposed only recently [2, 3], and their properties remain relatively unexplored. In this work, we have investigated two additional key aspects of these models: their noninvertible symmetries and their holographic interpretations. Through explicit analysis of several examples, we conclude that the presence of non-invertible symmetry and the corresponding bulk topological theory is an intrinsic feature of one-dimensional SPT phases protected by modulated symmetries, similar to those protected by non-modulated symmetries.

A subtlety remains in classifying various SSB phases dictated by a given symmetry group. As noted in all three examples of (c,d,e)SPTs and their KT-duals, the symmetry group of the dual phase is quite complex, involving some semidirect product structures and displaying several SSB models with the same symmetry group but different symmetry-breaking patterns. At present, these distinct SSB models are more sharply distinguished by their KT-duals, namely their corresponding SPT models. The SSB models and SPT models are in one-to-one correspondence, as the KT transformation becomes invertible when restricted to SPT states that are +1 eigenstates of the relevant global symmetries. While this observation suggests that NISPTs provide a useful diagnostic for how to distinguish various SSB phases in the presence of semidirect symmetry structure, it is conceivable that a more direct criterion that does not rely on gauging the symmetry might be developed to differentiate these SSB phases and define appropriate phase-specific invariants.

We remark that additional charge, dipole, exponential NIMSPT Hamiltonians may exist beyond those presented in this work. A systematic investigation of the classification of SPTs in the presence of non-invertible symmetry can be found, for instance, in [43, 47, 59]. The extension of these schemes to allow for the full classification of NIMSPTs remains an important direction for future study. As noted in [47], a SymTFT that fully captures non-invertible symmetries can, in principle, be obtained by gauging the appropriate anyon dualities within the SymTFT associated with invertible symmetries. This suggests that such a SymTFT could serve as a viable tool for the classification of NIMSPTs.

Finally, we note that the noninvertible symmetry of two-dimensional cluster state was recently analyzed in [70]. Whether their analysis of noninvertible symmetry in 2D cluster state can be extended to the dipolar cluster state recently proposed in [71] is an interesting problem.

## ACKNOWLEDGMENTS

We thank Ömer Aksoy, Gilyoung Cho, Ho Tat Lam, Yabo Li, Da-Chuan Lu, Aswin Parayil Mana, Salvatore Pace, Shu-Heng Shao, Zijian Song and Masahito Yamazaki for enlightening discussions. JHH thanks professor Yamazaki for hosting his visit to the University of Tokyo and professor Shu-Heng Shao for arranging a visit to MIT, and for enlightening discussions that took place at both institutions. J.H.H.
was supported by the National Research Foundation of Korea(NRF) grant funded by the Korea government(MSIT) (No. 2023R1A2C1002644). This research was finalized while visiting the Okinawa Institute of Science and Technology (OIST) through the Theoretical Sciences Visiting Program (TSVP). YY acknowledges support from NSF under award number DMR-2439118.

## Appendix A: Classification of SPT phases protected by two charge and two dipole symmetries

We summarize the SPT phases protected by two charge and two dipole symmetries [3]. The full classification yields $\mathbb{Z}_N^4$ distinct SPT phases, characterized by the parameters $(k_1, k_2, k_3, k_4)$. We focus on four representative cases: $(1,0,0,0)$, $(0,1,0,0)$, $(0,0,1,0)$, and $(0,0,0,1)$.

For $(k_1, k_2, k_3, k_4) = (1,0,0,0)$, the Hamiltonian is given by

$$H_d^{(1,0,0,0)} = -\sum_j Z_{2j-1} X_{2j} Z_{2j+1}^{-2} Z_{2j+3}$$
$$-\sum_j Z_{2j-2} Z_{2j}^{-2} X_{2j+1} Z_{2j+2} + \text{h.c.}. \quad \text{(A1)}$$

Within the decorated domain wall framework, this phase corresponds to the decoration of the charge operator of $C^o$ by the dipole domain wall operator of $D^e$, and of the charge operator of $C^e$ by the dipole domain wall operator of $D^o$. This results in projective representations between the edge operators associated with the fractionalization of $C^o$ and $D^e$, and of $C^e$ and $D^o$.

For $(k_1, k_2, k_3, k_4) = (0,1,0,0)$, the Hamiltonian is given by

$$H_d^{(0,1,0,0)} = -\sum_j Z_{2j-2} Z_{2j}^\dagger X_{2j} Z_{2j}^\dagger Z_{2j+2} - \sum_j X_{2j+1} + \text{h.c.}. \quad \text{(A2)}$$

This phase corresponds to the decoration of the charge operator of $C^o$ by the dipole domain wall operator of $D^o$, leading to the projective representation between the edge operators of $C^o$ and $D^o$.

Similarly, for $(k_1, k_2, k_3, k_4) = (0,0,1,0)$, the Hamiltonian takes the form

$$H_d^{(0,0,1,0)} = -\sum_j X_{2j} - \sum_j Z_{2j-1} Z_{2j+1}^\dagger X_{2j+1} Z_{2j+1}^\dagger Z_{2j+3} + \text{h.c.}, \quad \text{(A3)}$$

where the role of $C^o, D^o$ in (A2) is replaced by $C^e, D^e$.

Finally, for $(k_1, k_2, k_3, k_4) = (0,0,0,1)$, the Hamiltonian is given by

$$H_d^{(0,0,0,1)} = -\sum_j Z_{2j-3} Z_{2j-1}^{-3} X_{2j} Z_{2j+1}^3 Z_{2j+3}^\dagger$$
$$-\sum_j Z_{2j-2}^\dagger Z_{2j}^3 X_{2j+1} Z_{2j+2}^{-3} Z_{2j+4} + \text{h.c.}. \quad \text{(A4)}$$

Here, the charge operator of $C^o$ ($C^e$) is decorated by the quadrupole domain wall operator of $Q^e$ ($Q^o$), where $Q^e$ and

$Q^o$ are defined as

$$Q^o = \prod_j X_{2j+1}^{j^2}, \qquad Q^e = \prod_j X_{2j}^{j^2}. \qquad (A5)$$

This results in projective representations between the edge operators associated with $D^o$ and $D^e$.

## Appendix B: Interfacing $H_{c'}(\alpha)$ and $H_{c'}(\alpha')$

Consider the interfacial Hamiltonian given by:

$$H_{c'(\alpha)|c'(\alpha')} = -\sum_{j=1}^{l/2} \omega^{-\alpha} Z_{2j-1}^\dagger X_{2j-1} Z_{2j} - \sum_{j=1}^{l/2-1} Y_{2j-1} X_{2j} Y_{2j+1}^\dagger \left(1 + Z_{2j-2} X_{2j-1}^\dagger Z_{2j}^{-2} X_{2j+1} Z_{2j+2}\right)$$

$$- \sum_{j=l/2+1}^{L/2} \omega^{-\alpha'} Z_{2j-1}^\dagger X_{2j-1} Z_{2j} - \sum_{j=l/2+1}^{L/2-1} Y_{2j-1} X_{2j} Y_{2j+1}^\dagger \left(1 + Z_{2j-2} X_{2j-1}^\dagger Z_{2j}^{-2} X_{2j+1} Z_{2j+2}\right) + \text{h.c.}, \qquad (B1)$$

where $L$ and $l$ are multiples of $2N$. This Hamiltonian can be interpreted as $H_{c'}(\alpha)$ defined on $j=1$ to $l$ and $H_{c'}(\alpha')$ defined on $j=l+1$ to $j=L$. The ground state(s) of $H_{c'(\alpha)|c'(\alpha')}$ should satisfy

$$Z_{2j-2}^\dagger X_{2j-1} Z_{2j} = \omega^\alpha \quad (j=1,2,\cdots,l/2)$$
$$Y_{2j-1} X_{2j} Y_{2j+1}^\dagger = 1 \quad (j=1,2,\cdots,l/2-1)$$
$$Z_{2j-2}^\dagger X_{2j-1} Z_{2j} = \omega^{\alpha'} \quad (j=l/2+1,l/2+2,\cdots,L/2)$$
$$Y_{2j-1} X_{2j} Y_{2j+1}^\dagger = 1 \quad (j=l/2+1,l/2+2,\cdots,L/2-1)$$
$$(B2)$$

One can derive

$$C^o|\psi\rangle = |\psi\rangle, \qquad C^e|\psi\rangle = \tilde{X}_l \tilde{X}_L |\psi\rangle$$
$$Z_l \tilde{X}_l = \omega \tilde{X}_l \tilde{Z}_l, \qquad Z_L \tilde{X}_L = \omega \tilde{X}_L \tilde{Z}_L. \qquad (B3)$$

where two edge operators are

$$\tilde{X}_l = Y_{l-1} X_l Y_{l+1}^\dagger \qquad \tilde{X}_L = Y_{L-1} X_L Y_1^\dagger. \qquad (B4)$$

The action of $K_c$ within the ground state subspace can found by examining how $K_c$ acts on the operators in (2.58) within the ground state subspace [22]:

$$\tilde{X}_l K_c |\psi\rangle = \omega^{\alpha-\alpha'} K_c \tilde{X}_L^\dagger |\psi\rangle$$
$$\tilde{X}_L K_c |\psi\rangle = \omega^{\alpha'-\alpha} K_c \tilde{X}_L^\dagger |\psi\rangle$$
$$\tilde{Z}_l^\dagger \tilde{Z}_L K_c |\psi\rangle = K_c \tilde{Z}_l \tilde{Z}_L^\dagger |\psi\rangle$$
$$K_c^2 |\psi\rangle = N \sum_{k=1}^N (\tilde{X}_l \tilde{X}_L)^k |\psi\rangle. \qquad (B5)$$

An explicit expression for $K_c$ that satisfies these relations is given by:

$$K_c |\psi\rangle = \varepsilon P Z_l^{\alpha-\alpha'} Z_L^{\alpha'-\alpha} \sum_{k=1}^N (\tilde{X}_L \tilde{X}_R)^k |\psi\rangle$$

$$= \varepsilon P \sum_{k=1}^N K_{c,l}^{(k)} K_{c,L}^{(k)} |\psi\rangle, \qquad (B6)$$

where we introduce the abbreviation $K_{c,l}^{(k)} = Z_l^{\alpha-\alpha'} \tilde{X}_L^k$, $K_{c,L}^{(k)} = Z_L^{\alpha'-\alpha} \tilde{X}_R^k$ in the second line. The prefactor $\varepsilon$ is either +1 or -1, but determining its exact value requires laborious calculations that do not affect the outcome of the following discussion.

Finally, we can show that

$$K_{c,l}^{(k)} \tilde{X}_l = \omega^{\alpha-\alpha'} \tilde{X}_l K_{c,l}^{(k)}, \qquad K_{c,L}^{(k)} \tilde{X}_L = \omega^{\alpha'-\alpha} \tilde{X}_L K_{c,L}^{(k)} \qquad (B7)$$

and confirm that they form projective representations when $\alpha \neq \alpha'$. Therefore, we can conclude that $H_{c'}$ with different $\alpha$ lies in different SPT phases.

## Appendix C: Interfacing $H_c$ and $H_{c''}(\alpha)$

Now we introduce the following interfacial Hamiltonian:

$$H_{c|c''(\alpha)} = -\sum_{j=1}^{l/2-1} Z_{2j-2}^\dagger X_{2j-1} Z_{2j} - \sum_{j=0}^{l/2-1} Z_{2j-1} X_{2j} Z_{2j+1}^\dagger$$

$$- \sum_{j=l/2}^{L/2-1} \omega^{-\alpha} Z_{2j-1} X_{2j} Z_{2j+1}^\dagger$$

$$- \sum_{j=l/2+1}^{L/2-1} Y_{2j-2}^\dagger X_{2j-1} Y_{2j} (1 + Z_{2j-3}^\dagger X_{2j-2}^\dagger Z_{2j-1}^2 X_{2j} Z_{2j+1}^\dagger)$$

$$+ \text{h.c.}, \qquad (C1)$$

which is a model where $H_c$ occupies $0 \leq j \leq l-1$ ($l=$even) sites and $H_{c''}$ occupies $l \leq j \leq L-1$ sites, with $L-l$ chosen as a multiple of $2N$. Periodic boundary condition is assumed. The ground states should satisfy

$$Z_{2j-2}^\dagger X_{2j-1} Z_{2j} = 1 \quad (1 \leq j \leq l/2-1)$$
$$Z_{2j-1} X_{2j} Z_{2j+1}^\dagger = 1 \quad (0 \leq j \leq l/2-1)$$
$$Y_{2j-2}^\dagger X_{2j-1} Y_{2j} = 1 \quad (l/2+1 \leq j \leq L/2-1)$$
$$Z_{2j-1} X_{2j} Z_{2j+1}^\dagger = \omega^\alpha \quad (l/2 \leq j \leq L/2-1). \qquad (C2)$$

From these conditions, one can derive

$$C^o|\psi\rangle = \tilde{X}_{l-1}\tilde{X}_{L-1}|\psi\rangle, \qquad C^e|\psi\rangle = |\psi\rangle,$$
$$Z_{l-1}\tilde{X}_{l-1} = \omega\tilde{X}_{l-1}\tilde{Z}_{l-1}, \qquad Z_{L-1}\tilde{X}_{L-1} = \omega\tilde{X}_{L-1}\tilde{Z}_{L-1}, \quad \text{(C3)}$$

where

$$\tilde{X}_{l-1} = Z_{l-2}^{\dagger}X_{l-1}Y_l \qquad \tilde{X}_{L-1} = Y_{L-2}^{\dagger}X_{L-1}Z_L. \qquad \text{(C4)}$$

Using the following relations:

$$\tilde{X}_{l-1}K_c|\psi\rangle = \omega^{-\alpha}K_c\tilde{X}_{l-1}|\psi\rangle$$
$$\tilde{X}_{L-1}K_c|\psi\rangle = \omega^{\alpha}K_c\tilde{X}_{L-1}|\psi\rangle$$
$$Z_{l-1}^{\dagger}Z_{L-1}K_c|\psi\rangle = K_cZ_{l-1}^{\dagger}Z_{L-1}|\psi\rangle$$
$$K_c^2|\psi\rangle = N\sum_{k=1}^{N}(\tilde{X}_{l-1}\tilde{X}_{L-1})^k|\psi\rangle, \qquad \text{(C5)}$$

an explicit expression for $K_c$ becomes:

$$K_c|\psi\rangle = \varepsilon Z_{l-1}^{\alpha}Z_{L-1}^{-\alpha}\left(\sum_{k=1}^{N}(\tilde{X}_{l-1}\tilde{X}_{L-1})^k\right)|\psi\rangle$$
$$= \varepsilon P\left(\sum_{k=1}^{N}K_{c,l-1}^{(k)}K_{c,L-1}^{(k)}\right)|\psi\rangle, \qquad \text{(C6)}$$

where we introduce the abbreviations

$$K_{c,l-1}^{(k)} = Z_{l-1}^{\alpha}\tilde{X}_l^k, \quad K_{c,L-1}^{(k)} = Z_{L-1}^{-\alpha}\tilde{X}_L^k$$

in the second line. The prefactor $\varepsilon$ is either +1 or -1, but determining its exact value requires laborious calculations that do not affect the outcome of the ensuing discussion.

Finally, we can show that these fractionalized KW operators satisfy the algebra

$$K_{c,l-1}^{(k)}\tilde{X}_{l-1} = \omega^{\alpha}\tilde{X}_{l-1}K_{c,l-1}^{(k)}, \quad K_{c,L-1}^{(k)}\tilde{X}_{L-1} = \omega^{-\alpha}\tilde{X}_{L-1}K_{c,L-1}^{(k)}$$
$$\text{(C7)}$$

and confirm that they form projective representations for all $\alpha \neq 0$. Therefore, we can also conclude that $H_{c''}$ with $\alpha = 0$ lies in the same SPT phase as the cluster state and $H_{c''}$ with $\alpha \neq 0$ realizes a new SPT distinct from the cluster state, though $H_c$ and $H_{c''}$ are protected by the same symmetry group. Additionally, under the symmetry group generated by $C^o$ and $C^e$ alone, the Hamiltonians $H_c$ and $H_{c''}$ cannot be shown to represent different SPT phases.

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
