# Peer review of "Noninvertible symmetry and topological holography for modulated SPT in one dimension"

_SciPost Physics, doi:SciPost Phys. 19, 110 (2025)_

## Round 1 · Referee Report · Anonymous (Referee 2) · 2025-8-21

Report

The authors study one-dimensional models with various modulated symmetries and demonstrate that these models possess additional non-invertible symmetries. By applying the KT transformation, they map SPT models to SSB models and subsequently construct new SSB models with the same set of symmetries. Through dualization of these SSB models, they obtain SPT models whose distinctions from the original SPT models can be identified via boundary symmetry fractionalizations. Moreover, their bulk Hamiltonians are formulated in the spirit of Topological Holography.

The manuscript provides detailed, step-by-step derivations of the results, which have the potential to lead to important physical insights. Since the field of generalized symmetries is rapidly developing, shedding light on concrete examples is highly valuable at this stage. Nevertheless, I believe there is room for further improvement in the manuscript, and I list my requested changes below.

1-Is the Hamiltonian (2.8) originally written by the authors? Otherwise, I encourage the authors to cite proper references around (2.8).

2-The authors introduce a new SSB model (2.31) and claim that "$\alpha \in \mathbb{Z}_N$ characterizes a distinct SPT". Though I found that the authors apply a KT transformation to get an SPT model (2.36), it could be misleading in that readers may understand (2.31) as a new SPT model. I recommend the authors to add some sentences to clarify this.

3-I personally found that the authors' detailed calculation is helpful to understand the content of the manuscript, but some readers might be overwhelmed by hundreds of equations. It would be better if the authors could include a section right after Introduction that summarizes main results with selected equations.

4-As the authors point out in the last paragraph of Summary and Discussion, the different SSB phases proposed in the manuscript are only distinguished by their KT duals. To claim this, it is needed to be ensured that the KT transformation is injective. I wonder if it has been proven and encourage the authors to include remarks on this.

5-As the authors write in Summary and Discussion, it seems that the authors try to claim that the presence of noninvertible symmetry and a corresponding bulk topological theory is a special property of modulated symmetries. However, the claim might be inappropriate and should be revised because there are examples (e.g. the standard transverse-field Ising model) where non-modulated symmetries accompany KW self-duality, which can be interpreted as noninvertible symmetries. I recommend the authors to revise the paragraphs to avoid confusion.

Recommendation

Ask for minor revision

---

## Round 1 · Referee Report · Anonymous (Referee 1) · 2025-8-21

Report

The authors study the symmetry protected topological (SPT) phases protected by non-invertible symmetries that include modulated symmetries. By way of examples, the authors construct SPT ground states of dipole and exponential modulated symmetries together with an appropriate non-anomalous non-invertible self-duality symmetry.

The paper is well-written with many details explicitly shown. I found the results interesting and significant enough to warrant a publication at Scipost Physics. However, I think the manuscript would benefit from a revision according to the following points.

1) The authors start with reviewing what they call "charge SPT" which is the generalization of the well-known cluster model to that with $\mathbb{Z}_N\times\mathbb{Z}_N$ symmetry. However, the symmetry here is a "charge" symmetry only if one combines the even and odd sites into larger unit cells supporting two copies of $\mathbb{Z}_N$ degrees of freedom. In particular, the symmetry operators $\prod X$ on even and odd sites are also modulated with the modulating function being $f_j = (j \text{ mod } 2)$. Therefore, I suggest reviewing this section to justify why these symmetries are "charge" and not modulated (perhaps by enlarging the unit cells so that both symmetry generators are translationally invariant).

2) Related to the above point, an important property of the modulated symmetries is that the translation symmetry (more generally spatial symmetries) has a non-trivial action on such internal symmetries. I think the lack of emphasis and discussion on this point weakens the point authors are trying to make. In particular, if we completely disregard spatial symmetries then all the examples in the manuscript can be recast into ordinary internal symmetries by appropriately enlarging the unit cells. This renders the adjactive "modulated" unnecessary. I suggest authors to incorporate translation (or reflection) symmetries into the discussion in all sections.

3) While the authors give examples of SPT states with non-invertible symmetries, the corresponding "holographic description" is only for the invertible part of the symmetries. This leads to two questions to be addressed. First, can the authors comment on what would be the bulk topological order/topological field theory (TFT) that would describe the non-invertible modulated SPTs fully? Such TFTs should be obtained by gauging in the bulk the automorphisms (such as layer swap) that correspond to the non-invertible symmetry in the boundary. Second, related to the comment 2), what is the role of the spatial symmetries in the symTFT description so that we know the boundary theory has indeed a modulated symmetry?

4) Do the authors know if their examples of noninvertible SPTs are in any sense complete? In other words, do the authors know the number of fiber functors of the fusion category smmetry protecting such non-invertible SPTs. It would be interesting to see if the families of Hamiltonians presented here exhaust all possible SPT phases with the appropriate fusion category symmetry.

Recommendation

Ask for minor revision

---

## Round 2 · Author Response

Warnings issued while processing user-supplied markup:
- Inconsistency: plain/Markdown and reStructuredText syntaxes are mixed. Markdown will be used.
Add "#coerce:reST" or "#coerce:plain" as the first line of your text to force reStructuredText or no markup.
You may also contact the helpdesk if the formatting is incorrect and you are unable to edit your text.
Dear editors,
We thank you for sending us the comments of two reviewers on our manuscript “Noninvertible symmetry and topological holography for modulated SPT in one dimension”. We have made extensive revisions to the manuscript in faithfully addressing the reviewers’ comments. Please see also our responses to various portions of the referee comments.
yours sincerely, Jintae Kim, Yizhi You, and Jung Hoon Han
Report #1
The authors study the symmetry protected topological (SPT) phases protected by non-invertible symmetries that include modulated symmetries. By way of examples, the authors construct SPT ground states of dipole and exponential modulated symmetries together with an appropriate non-anomalous non-invertible self-duality symmetry. The paper is well-written with many details explicitly shown. I found the results interesting and significant enough to warrant a publication at Scipost Physics. However, I think the manuscript would benefit from a revision according to the following points.
Comment: The authors start with reviewing what they call "charge SPT" which is the generalization of the well-known cluster model to that with Z_N \times Z_N symmetry. However, the symmetry here is a "charge" symmetry only if one combines the even and odd sites into larger unit cells supporting two copies of Z_N degrees of freedom. In particular, the symmetry operators \prod_X on even and odd sites are also modulated with the modulating function being f_j= (j mod 2). Therefore, I suggest reviewing this section to justify why these symmetries are "charge" and not modulated (perhaps by enlarging the unit cells so that both symmetry generators are translationally invariant).
Reply: Indeed, the two “charge” symmetries of the cluster model are not ordinary global symmetries in the sense that the relevant symmetry operators are invariant under translation symmetry under two-site translation, not one-site translation, since we treat even vs odd sites as two sublattices. We note this point and add a qualifying comment to this effect in the revised manuscript to more carefully justify the use of the term “charge” symmetry.
Comment: Related to the above point, an important property of the modulated symmetries is that the translation symmetry (more generally spatial symmetries) has a non-trivial action on such internal symmetries. I think the lack of emphasis and discussion on this point weakens the point authors are trying to make. In particular, if we completely disregard spatial symmetries then all the examples in the manuscript can be recast into ordinary internal symmetries by appropriately enlarging the unit cells. This renders the adjective "modulated" unnecessary. I suggest authors to incorporate translation (or reflection) symmetries into the discussion in all sections.
Reply: This comment is also essential for our manuscript. We have added a rigorous definition of modulated symmetries in the beginning of Sec.II, characterized by their non-trivial transformation under elements of the spatial symmetry group, and have noted that dipole and exponential symmetries provide concrete examples. Furthermore, as suggested by the referee, we have incorporated discussions of translational symmetries in other sections to provide a clearer explanation. (Added discussions are highlighted in red for visibility.) In addition, we would like to note that for the exponential Z_N-symmetric SPT discussed in Sec.V, when a and N are coprime, it is not possible to enlarge the lattice and reinterpret the symmetry as an ordinary internal one. This is because the charge oscillates in a nonperiodic manner. (Indeed, for coprime $ and N, there is no integer m such that a^m =1 \mod{N}.) Thus, not all modulated symmetries can be incorporated into ordinary internal symmetries on a superlattice.
Comment: While the authors give examples of SPT states with non-invertible symmetries, the corresponding "holographic description" is only for the invertible part of the symmetries. This leads to two questions to be addressed. First, can the authors comment on what would be the bulk topological order/topological field theory (TFT) that would describe the non-invertible modulated SPTs fully? Such TFTs should be obtained by gauging in the bulk the automorphisms (such as layer swap) that correspond to the non-invertible symmetry in the boundary.
Reply: From the SymTFT perspective, when the symmetry group consists solely of invertible symmetries, the associated non-invertible symmetries emerge via the nontrivial anyonic symmetries of the SymTFT, such as the electric-magnetic anyon duality in the toric code. It means we can implement the non-invertible symmetries by changing the boundary condition. However, in the presence of multiple non-invertible symmetries, all such symmetries can, in principle, be realized on the boundary. Consequently, isolating and attributing a specific non-invertible symmetry to the boundary becomes ambiguous.
As the referee correctly pointed out, a SymTFT that fully captures non-invertible modulated SPT phases can, in principle, be constructed by gauging the specific anyon duality in the SymTFT for invertible symmetries. However, our primary goal was to demonstrate that the Hamiltonians we constructed are invariant under non-invertible symmetries, which we verified by appropriately modifying the boundary conditions. This approach suffices for our purposes.
We agree that a full treatment of non-invertible modulated SPT phases via bulk gauging of anyon duality is a promising direction, and would likely enable a complete classification of such phases. We have included this point in the discussion section as a potential avenue for future work.
Comment: Second, related to the comment 2), what is the role of the spatial symmetries in the symTFT description so that we know the boundary theory has indeed a modulated symmetry?
Reply: We also comment in the revised manuscript that translational symmetries in the symTFT framework play a significant role, as certain Wilson loop operators (Eqs.(3.53), (3.60), (4.57)) associated with the bulk topological order are modulated and exhibit nontrivial transformations under translation operations. In general, a modulated SPT corresponds to a bulk described by a modulated (higher-rank) gauge theory, where charge and flux excitations transform nontrivially under lattice translations.
Comment: Do the authors know if their examples of noninvertible SPTs are in any sense complete? In other words, do the authors know the number of fiber functors of the fusion category smmetry protecting such non-invertible SPTs. It would be interesting to see if the families of Hamiltonians presented here exhaust all possible SPT phases with the appropriate fusion category symmetry.
Reply: According to the paper by J. Maeda and T. Oishi, the number of charge NIMSPTs is $N^2 f(N)$ for odd $N$ and $\frac{3}{4} N^2 f(N)$ for even $N$, where $f(N)$ denotes the number of $k$ satisfying $\gcd(N,k)=d$ with $d^2 \equiv 0 (mod N)$. These results indicate that the set of charge NIMSPT examples presented here is not exhaustive, and we expect a similar situation for dipole and exponential NIMSPTs. We mentioned these results in the charge NIMSPT section. Additionally, in the Summary and Discussion section, we emphasized this point and proposed the classification of NIMSPTs as an important direction for future research, in connection with recent advances in the classification of MSPTs.
Report #2
The authors study one-dimensional models with various modulated symmetries and demonstrate that these models possess additional non-invertible symmetries. By applying the KT transformation, they map SPT models to SSB models and subsequently construct new SSB models with the same set of symmetries. Through dualization of these SSB models, they obtain SPT models whose distinctions from the original SPT models can be identified via boundary symmetry fractionalizations. Moreover, their bulk Hamiltonians are formulated in the spirit of Topological Holography.
The manuscript provides detailed, step-by-step derivations of the results, which have the potential to lead to important physical insights. Since the field of generalized symmetries is rapidly developing, shedding light on concrete examples is highly valuable at this stage. Nevertheless, I believe there is room for further improvement in the manuscript, and I list my requested changes below.
Comment: Is the Hamiltonian (2.8) originally written by the authors? Otherwise, I encourage the authors to cite proper references around (2.8).
Reply: We added references for (2.8).
Comment: The authors introduce a new SSB model (2.31) and claim that "α∈Z_N characterizes a distinct SPT". Though I found that the authors apply a KT transformation to get an SPT model (2.36), it could be misleading in that readers may understand (2.31) as a new SPT model. I recommend the authors to add some sentences to clarify this.
Reply: Thank you for noting the typo. We addressed the sentence that α∈Z_N characterizes a distinct SSB since here (2.31) represents SSBs.
Comment: I personally found that the authors' detailed calculation is helpful to understand the content of the manuscript, but some readers might be overwhelmed by hundreds of equations. It would be better if the authors could include a section right after Introduction that summarizes main results with selected equations.
Reply: For readers, we have added a summary of the NIMSPTs as a subsection in the Summary and Discussion section.
Comment: As the authors point out in the last paragraph of Summary and Discussion, the different SSB phases proposed in the manuscript are only distinguished by their KT duals. To claim this, it is needed to be ensured that the KT transformation is injective. I wonder if it has been proven and encourage the authors to include remarks on this.
Reply: KT transformations are generally non-injective; however, when acting on states that are +1 eigenstates of certain global symmetries, they behave effectively as unitary operators (up to normalization). See Eqs. (2.16), (3.14), (4.15) and the following explanation. This implies that the KT transformation becomes bijective (invertible) when restricted to such a subspace. Since NIMSPTs are +1 eigenstates of the relevant global symmetries, their KT duals—corresponding to distinct SSB phases—are in one-to-one correspondence. We have noted this point explicitly in Summary and DIscussion.
Comment: As the authors write in Summary and Discussion, it seems that the authors try to claim that the presence of noninvertible symmetry and a corresponding bulk topological theory is a special property of modulated symmetries. However, the claim might be inappropriate and should be revised because there are examples (e.g. the standard transverse-field Ising model) where non-modulated symmetries accompany KW self-duality, which can be interpreted as noninvertible symmetries. I recommend the authors to revise the paragraphs to avoid confusion.
Reply: We agreed that the presence of non-invertible symmetry and a corresponding bulk topological theory is a property of both non-modulated and modulated symmetries. To avoid confusion, we revised the sentence to “Through explicit analysis of several examples, we conclude that the presence of non-invertible symmetry and a corresponding bulk topological theory is also a feature of one-dimensional SPT phases protected by modulated symmetries, similar to those protected by non-modulated symmetries.

---

## Round 2 · List of Changes

- Edited the definition of modulated symmetries from the perspective of spatial symmetries
- Added explanations of modulated symmetries associated with translational symmetries in several sections
- Added a "Summary of our findings" subsection in the Summary and Discussion section.
- Corrected typos.
- Added references.

---

## Editorial Decision

published